# Exploring Components, Sensors, and Techniques for Cancer Detection via eNose Technology: A Systematic Review

**DOI:** 10.3390/s24237868

**Published:** 2024-12-09

**Authors:** Washington Ramírez, Verónica Pillajo, Eileen Ramírez, Ibeth Manzano, Doris Meza

**Affiliations:** 1Departamento de Ciencias de la Computación, Universidad de las Fuerzas Armadas ESPE, Av. Gral. Rumiñahui S/N, Sangolquí 171104, Ecuador; iamanzano@espe.edu.ec; 2Departamento de Informática, Universidad Politécnica Salesiana, Quito 170146, Ecuador; vpillajon@est.ups.edu.ec; 3Facultad de Medicina, Pontificia Universidad Católica del Ecuador, Quito 170143, Ecuador; 4Facultad de Ciencias Económicas, Universidad Central del Ecuador, Quito 170521, Ecuador; dmeza@uce.edu.ec

**Keywords:** cancer detection, eNose, biomarkers, components, sensors, machine learning techniques, medical devices, Support Vector Machine, Principal Component Analysis, systematic review

## Abstract

This paper offers a systematic review of advancements in electronic nose technologies for early cancer detection with a particular focus on the detection and analysis of volatile organic compounds present in biomarkers such as breath, urine, saliva, and blood. Our objective is to comprehensively explore how these biomarkers can serve as early indicators of various cancers, enhancing diagnostic precision and reducing invasiveness. A total of 120 studies published between 2018 and 2023 were examined through systematic mapping and literature review methodologies, employing the PICOS (Population, Intervention, Comparison, Outcome, and Study design) methodology to guide the analysis. Of these studies, 65.83% were ranked in Q1 journals, illustrating the scientific rigor of the included research. Our review synthesizes both technical and clinical perspectives, evaluating sensor-based devices such as gas chromatography–mass spectrometry and selected ion flow tube–mass spectrometry with reported incidences of 30 and 8 studies, respectively. Key analytical techniques including Support Vector Machine, Principal Component Analysis, and Artificial Neural Networks were identified as the most prevalent, appearing in 22, 24, and 13 studies, respectively. While substantial improvements in detection accuracy and sensitivity are noted, significant challenges persist in sensor optimization, data integration, and adaptation into clinical settings. This comprehensive analysis bridges existing research gaps and lays a foundation for the development of non-invasive diagnostic devices. By refining detection technologies and advancing clinical applications, this work has the potential to transform cancer diagnostics, offering higher precision and reduced reliance on invasive procedures. Our aim is to provide a robust knowledge base for researchers at all experience levels, presenting insights on sensor capabilities, metrics, analytical methodologies, and the transformative impact of emerging electronic nose technologies in clinical practice.

## 1. Introduction

Early cancer detection remains one of the greatest challenges in modern medicine. Traditionally, the diagnosis of this disease has relied on invasive and costly techniques, such as biopsies and imaging scans, which require complex procedures and may pose risks to patients. However, a recent innovation has opened up new possibilities: Volatile Organic Compounds (VOCs). These compounds, present in biomarkers such as breath, saliva, urine, and blood, offer a non-invasive indication of the body’s physiological processes and may be associated with various forms of cancer [1]. The rapid advancement of eNose technology offers transformative potential in cancer diagnostics by leveraging VOCs as biomarkers. This study systematically explores key components, sensor technologies, and analytical techniques to enhance the precision and accessibility of early cancer detection.

VOCs are generated in small concentrations during altered metabolic processes, becoming a powerful tool for early cancer identification. The accurate measurement of these compounds through advanced technologies has shown great potential in the non-invasive detection of diseases, which could revolutionize traditional diagnostic approaches. This ability to detect VOCs may enable faster, more accessible, and less invasive diagnoses, significantly changing the way complex diseases like cancer are diagnosed.

In this context, electronic nose (eNose) technology has emerged as a revolutionary tool for early cancer detection. By mimicking the human sense of smell, the eNose can identify specific VOC patterns associated with various types of cancer. This technology uses gas sensor arrays to detect and analyze these volatile compounds. Over recent years, the eNose has evolved rapidly, from being an experimental tool to becoming a promising alternative for identifying specific odors present in biological biomarkers. This approach not only offers the possibility of faster diagnoses, but it could also facilitate the implementation of non-invasive methods in medical diagnostics. Despite the progress made, the clinical implementation of eNose devices still faces several challenges, such as sensor optimization, data integration, and adapting the technology to real-world medical environments. In response to this knowledge gap, this study proposes to conduct a comprehensive analysis using systematic mapping (SM), a systematic literature review (SLR), and PICOS methodologies [2,3] with the aim of gathering and analyzing key information regarding the detection of VOCs associated with cancer, addressing their chemical components, concentrations, techniques, and employed metrics.

This study aims to fill some of the current gaps in understanding how VOCs can serve as biomarkers for early cancer detection, laying the foundation for the development of a non-invasive diagnostic tool. Through this innovative approach, it is expected to contribute to scientific advances in early cancer detection and improve medical diagnostic and treatment options. By reducing dependence on invasive methods, this research has the potential to enhance patients’ quality of life and provide more accessible and accurate diagnoses in clinical practice.

The reliability of detections made by eNose devices for cancer biomarkers has been extensively analyzed in terms of performance metrics such as sensitivity (SE) and specificity (SP). Sensitivity measures the ability of the device to correctly identify individuals with the disease (true positives), whereas specificity indicates its ability to correctly identify those without the disease (true negatives). Table 1 compares eNose detection metrics with traditional methods across various cancer types, demonstrating comparable and promising results.

This comparative analysis indicates that both traditional diagnostic methods and eNose detection provide acceptable levels of reliability, as evidenced by the reported sensitivity and specificity percentages for various cancer types. eNose detection emerges as a promising and favorable alternative, particularly due to its non-invasive and potentially accessible nature, making it an attractive tool for early cancer detection. However, despite its advances and encouraging results, it is important to emphasize that eNose detection should be supplemented with traditional techniques, such as biopsies or imaging diagnostics, to confirm diagnoses and achieve definitive results. While eNose detection does not replace traditional methods, its development and application represent a significant step toward creating more accessible and less invasive diagnostic tools. The ability to efficiently and non-invasively identify cancer biomarkers opens new opportunities for screening and patient monitoring, thereby enhancing early detection and clinical outcomes.

## 2. Background

### 2.1. Volatile Organic Compounds

Volatile Organic Compounds (VOCs) are a valuable source of information about an individual’s physiological and metabolic status. They can be generated and released through various means, such as breath, blood, skin, urine, and feces [104]. VOCs are significant in identifying various compounds associated with different types of cancer, which underscores their relevance in disease diagnosis. They provide a non-invasive approach that contributes to early identification and effective treatment. It is noteworthy that VOCs are not only present in the human body but are also found in the environment. Prolonged exposure to these environmental compounds, originating from sources such as industrial chemicals, paints, and solvents [16], can have significant repercussions on health, causing respiratory problems and affecting overall well-being. Therefore, understanding and controlling both endogenous emission and exogenous exposure to VOCs are essential for maintaining human health. To measure cancer-generated compound concentrations, various units are employed, such as ng/μL, mass/charge ratios (*m*/*z*), and mg/L [27]. These units provide crucial information about the relative quantity of specific substances, facilitating the accurate assessment of cancer-associated compounds in diverse contexts such as biological biomarkers or clinical environments. Concentration measurement is fundamental to understanding the relevance and quantity of present VOCs, enabling a more precise interpretation of research results, and contributing to advancements in understanding their role in human health.

### 2.2. Cancer

Cancer, a deadly disease, presents itself in various forms, characterized by the uncontrolled proliferation of abnormal cells in different tissues of the body, as documented in recent research [121]. Globally, statistics reveal a significant burden, with estimates indicating approximately 19.3 million new cancer cases in the year 2020, highlighting its widespread impact. In this context, nearly 10 million deaths attributed to cancer were recorded, emphasizing the urgency of addressing this public health issue [123]. In the specific context of Ecuador, 29,273 new cancer cases were reported in the same year, resulting in 15,123 patient fatalities [123].

For this study, nine types of cancer were identified, with breast cancer being the most frequently diagnosed, representing 11.7% of the total cases. It is closely followed by lung cancer (11.4%), colorectal cancer (10.0%), prostate cancer (7.3%), stomach cancer (5.6%), cervical cancer (3.1%), bladder cancer (3.0%), pancreatic cancer (2.6%), and lip and oral cavity cancer (2.0%). Regarding mortality, lung cancer tops the list with 18.0%, which is followed by colorectal cancer (9.4%), stomach cancer (7.7%), breast cancer (6.9%), pancreatic cancer (4.7%), prostate cancer (3.8%), cervical cancer (3.4%), bladder cancer (2.1%), and lip and oral cavity cancer (1.8%) [123]. Lung cancer begins in the normal bronchial epithelium and progresses from dysplasia to carcinoma, eventually advancing to invasive cancer. This transformation involves the activation of oncogenes, inactivation of tumor suppressor genes, and genomic instability [124]. To detect and diagnose lung cancer, mass spectrometry is used to measure ion mass, while gas chromatography is employed to separate and analyze VOCs. Gastric cancer, also known as stomach cancer, arises in the lymphoid tissue of the gastric mucosa and submucosa due to genetic changes in the cells lining the stomach, leading to uncontrolled growth. The strategies used for its detection and diagnosis include endoscopies, biopsies, and imaging studies to assess tumor extension [125]. The treatment of gastric cancer varies according to the cancer stage and characteristics and includes surgery, chemotherapy, radiotherapy, and targeted therapies. Cervical cancer is caused by the human papillomavirus (HPV), with HPV 16 and 18 being the most dangerous strains [126]. To detect this cancer, medical professionals use tools such as pap smear tests, HPV DNA detection tests, and colposcopies. Colorectal cancer originates from adenomatous polyps [127]. Physicians use colorectaloscopies, fecal occult blood tests, sigmoidoscopies, and computed tomography colorectalography to detect and diagnose this cancer. Prostate cancer is characterized by the formation of adenocarcinomas and carcinomas in various areas of the prostate gland [128]. To detect this cancer, medical professionals use the prostate-specific antigen blood test, digital rectal examination, and prostate biopsy. Pancreatic cancer is a term used to describe various tumors such as pancreatic ductal adenocarcinoma, cystadenocarcinomas, neuroendocrine tumors, pancreatic sarcoma, acinar cell carcinoma, and lymphoma [129]. To detect and diagnose pancreatic cancer, computed tomography and magnetic resonance imaging are used to visualize the anatomy and structure of the pancreas. Endoscopic retrograde cholangiopancreatography is also employed to evaluate the pancreatic duct system. Bladder cancer begins in the inner lining of the bladder and can be categorized as transitional cell carcinoma, urothelial carcinoma, squamous cell carcinoma, or adenocarcinoma [130]. To detect and diagnose bladder cancer, cystoscopy is used to directly visualize the bladder interior using a thin lighted tube. Imaging tests such as computed tomography and magnetic resonance imaging are used to assess the extent of the tumor. Urine analyses, particularly urine cytology, are used to identify cancer cells in biomarkers.

### 2.3. Medical Devices

Medical devices specifically designed for identifying VOCs have undergone significant advancements in detecting component concentrations, substantially contributing to the improvement of disease detection and prevention. These technological tools are crucial in the realm of public health, as they not only enable the identification of biomarkers and chemicals but also play a prominent role in exploring the detection of components associated with various types of cancer. The early identification of anomalous concentrations of specific biomarkers linked to different types of cancer has become a top priority in medical research. The enhanced sensitivity and specificity of these devices enable an earlier and more accurate detection of indicators that could suggest the presence of cancer cells. Furthermore, the ability to continuously monitor these biomarkers facilitates tracking the progression of the disease and evaluating the effectiveness of treatments [87]. The significance of thoroughly exploring these devices for the detection of cancer-related components lies in their potential to revolutionize medical care. By providing faster and more precise diagnoses, these advancements could significantly impact survival rates and the quality of life for patients [92].

### 2.4. Sensors

The detection of odors through specialized devices is an important area of research, as these devices can identify particular compounds. These sensors generate precise electrical signals by detecting the presence of specific gases in the environment [62]. Recent studies have highlighted their ability to recognize and respond to specific types of VOCs [25]. This makes them an invaluable tool in various applications, from identifying unpleasant odors in everyday environments to monitoring chemical emissions in industrial settings [43]. In this research, we will explore the importance of these devices and their unique ability to generate precise electrical signals in the presence of specific substances. Biomarkers such as blood, urine, saliva, and breath can be used to analyze an individual’s physiological state. Blood is collected through venipuncture [131], while urine is produced by the kidneys, which filter and eliminate waste products and excess substances from the body [132]. Saliva is formed through the activity of salivary glands [133], and breath biomarkers require specific instructions, such as refraining from eating or drinking specific foods and liquids for a specified time [72].

## 3. Research Method

### 3.1. Definition of the Research Question

The PICOS methodology was employed to structure the research questions and construct the search strings for scientific repositories, encompassing both technical studies (TS) and medical studies (MS). This methodology has proven effective in enhancing the precision of the search for relevant information on the research topic, through five components detailed in Table 2, which are crucial when formulating research questions [3]. In this study, questions were formulated and categorized as questions for systematic mapping (QSM) and questions for systematic literature review (QSR) for SM and SLR, respectively. These questions are detailed below.
QSM1: How do different systematic search methodologies affect the quality and quantity of data retrieved on volatile compounds related to early cancer detection, and what approaches optimize the precision of scientific data collection?QSM2: How has the publication of studies on volatile compound detection for early cancer identification evolved over the last five years, what are the emerging trends in the use of biomedical sensors such as eNose technology, and are there gaps in the literature that limit its development?QSM3: What are the primary studies and medical devices used in the detection of VOCs associated with cancer, and how have these devices evolved in terms of technological features and clinical relevance?QSR1: What are the most commonly used machine learning techniques and algorithms in the detection of VOCs associated with cancer, and how do they contribute to the accuracy and efficiency of cancer diagnosis using eNose technology?QSR2: What are the key VOCs identified as biomarkers for various types of cancer, and how do their concentrations and chemical classifications contribute to early detection and diagnosis?QSR3: What types of sensors have proven most effective for the detection of VOCs in cancer diagnostics, and what are the challenges and limitations in their implementation in clinical settings?QSR4: What are the performance metrics most commonly used to evaluate the effectiveness of eNose technology in detecting cancer-related VOCs, and how do these metrics vary across different studies and cancer types?PSR5: Are there any standardized databases or datasets specifically designed for VOCs detection in the context of cancer diagnosis, and how do these databases support the development of more accurate diagnostic tools?

The keywords and the combination of boolean operators, for both MS and TS, have resulted in the formulation of the main search strings. For technical studies, the string is composed of “Cancer AND Volatile Organic Compounds AND Techniques AND (Electronic Nose OR eNose OR Sensor)”. Meanwhile, the primary search string for medical studies took the form of “(Cancer OR Cancer Cell) AND Volatile Organic Compounds AND (Concentrations OR Biomarkers)”. These search strings were applied across six repositories: Scopus, Science Direct, Web of Science, PubMed, IEEE Xplore, Springer and ProQuest. In the process of selecting relevant medical and technical studies, inclusion criteria are employed to define the desirable characteristics. In parallel, exclusion criteria establish limits to discard those that do not meet these requirements, ensuring coherence and relevance in the systematic review process. In Table 3, three inclusion criteria are specified for systematic mapping (CiSM) and seven for systematic literature review (CiSR). Furthermore, three exclusion criteria have been defined for systematic mapping (CeSM) and three for systematic literature review (CeSR). To subsequently refine the search results and increase the probability of obtaining reliable information, inclusion and exclusion criteria were established based on specific conditions for the bibliographic review.

### 3.2. Execution

During this phase, we outline research questions, keywords, and related synonyms, as well as search strings for MS and TS compilation. For the purpose of systematic mapping, we applied three inclusion criteria and three exclusion criteria simultaneously along with the terms identified in PICOS. The criteria used in SM for the review of TS provided 2896 studies, while for MS, the figure was 3847 studies. These data were collected from six repositories, as detailed in Table 4. For the systematic literature review of TS, we employed six inclusion criteria and three exclusion criteria, which were organized into phases (F). In F1, we applied CeSR1 and CeSR2, resulting in a total of 796 excluded studies. F2 included CiSR1 and CiSR5, resulting in 340 studies. In F3, we included CiSR2 and CiSR3, resulting in a total of 133 studies. For F4, we included CiSR6 and CiSR7, totaling 60 studies. For systematic literature reviews of medical studies, we used seven inclusion criteria and three exclusion criteria organized into phases (F). In F1, we excluded CeSR1 and CeSR2, resulting in a total of 1180 excluded studies. In F2, we included CiSR1, CiSR4, and CiSR5, providing 438 studies. In F3, we included CiSR2 and CiSR3, resulting in a total of 230 studies. Finally, 60 studies were included in F4, CiSR6, and CiSR7. Table 4 concisely summarizes all of these details. The analysis produced 120 studies with an equal distribution of 60 in TS and 60 in MS. Figure 1 provides a visual summary of the publication distribution per year, showing that the majority of studies were conducted in 2021, contributing 30% of the total 120 publications. Furthermore, it is evident that the years 2018 and 2023 represent approximately 15% and 19.17%, respectively. Similarly, the years 2019, 2020, and 2022 made comparable contributions, each representing between 11.67% and 12.5% of the total studies. The analysis of the 120 publications shows that 65.83% belong to Q1 journals (79), followed by 25.83% in Q2 (31), 5% in Q3 (6) and 0.83% in Q4 (1). In addition, 2.5% (3) are conferences. This indicates a significant predominance of high-impact publications in Q1 with a lower representation of conferences within the distribution.

### 3.3. Taxonomy

The study encompasses 120 investigations that identify 20 categories of VOCs distributed across 9 types of cancers. These categories are organized visually in Figure 2. The components are extracted from urine, breath, saliva and blood biomarkers using 110 sensors and 22 medical devices. Additionally, they are analyzed through 34 techniques, including segmentation, feature extraction, correlation, regression, and classifiers.

## 4. Research Topics on VOCs for eNose

### 4.1. VOCs Databases

In this study, we have identified a total of 11 public databases and 47 private databases. It is worth noting that we have primarily focused on public databases, as highlighted in Figure 3. This is because they are more accessible and allow for relevant analysis that can be useful for future research.

The Human Metabolome Database (DBMP-A) (https://hmdb.ca/metabolites (accessed on 5 October 2024)) is a compilation of information on small molecule metabolites present in the human body. It consists of 220,945 entries and 130 data fields [134]. Sensors are employed to analyze the chemical structure, properties, and concentrations of the components on the technical side [122]. Data are categorized into groups, and features are extracted to form a test set, which is used to evaluate instrument measurement data through the eNose [93]. In the medical domain, ref. [77] utilizes the data to assess differences between groups and correlation coefficients among VOCs. Meanwhile, ref. [4] experiments with the data to detect and discriminate complex VOCs mixtures associated with diseases. Raw data from 174 biomarkers are extracted [72] and divided into two groups for comparing exhaled VOCs with the Human Metabolome Database DBMP-A. In [115], compound data are used to prepare a matrix for chemometric analysis and construct a dataset representing the distributions of 373 compounds in all biomarkers. In [65,85,117], experiments with the data are used to identify compounds and subsequently include them in a matrix for the corresponding analysis. Finally, in [86], data are extracted, population and neoplasia characteristics are collected, and the sampling and analysis methodology is conducted to standardize these biomarkers with the HMDB-A.

The NIST Database (DBMP-C) (https://chemdata.nist.gov/ (accessed on 5 October 2024)) is a specialized compound identification database that provides reference mass spectra for GC-MS (gas chromatography–mass spectrometry) and LC-MS (liquid chromatography–tandem mass spectrometry) [135].

Its utility in medical research has been demonstrated through the identification of compounds based on retention time, fragmentation patterns, and spectral matching with the NIST database. Quality assessment of the dataset involved analyzing a standard deviation exceeding 30%, as discussed in [7]. In [106], multivariate and univariate statistical tests were conducted using training and test sets. VOCs identification in [21,76] was accomplished by comparing chromatogram spectra with the NIST mass spectra library. In [79], NIST data were compared with a sample of 800 volatile metabolites, while in [109], candidate biomarkers were identified by comparing 46 cancer patient breath biomarkers with the information from the NIST database.

The Neomeditec Database (DBMP-F) (https://neomeditec.com/VOCdatabase/ (accessed on 5 October 2024)) serves as a comprehensive repository of information on 327 biomarkers, encompassing CAS numbers and their respective cancer associations. In [18], this database was pivotal in evaluating diverse VOCs and identifying promising biomarkers. Similarly, the Sigma Aldrich Alfa Aesar Database (DBMP-B) (https://www.chemspider.com/Search.aspx?dsn=Sigma-Aldrich (accessed on 26 November 2024)) includes detailed information on biomarkers, presenting their chemical formula, molecular weight, and identifiers. In [5], this database captured VOCs in static and dynamic environments. In the realm of technical literature [99,111], the eNose onboard database (DBTP-B) was utilized for initial data collection, reduction of principal components, and discriminant analysis to calculate precision percentages. Furthermore, ref. [80] utilized this database to visualize sensor signals and identify those exhibiting significantly elevated responses. In [34,39], an eNose system, equipped with MOS (Semiconductor Metal Oxide Gas) gas sensors, was applied to estimate high-dimensional data, classify, and categorize components.

### 4.2. Analysis of Volatile Organic Compounds

To familiarize the reader with the labeling system used in this study, **H** and $H_{2}$ are employed as illustrative examples. Although **H** and $H_{2}$ represent the same element, hydrogen, they exhibit significant differences in their chemical properties and uses. Atomic hydrogen (**H**) is highly reactive and primarily used in specific scientific research. In contrast, molecular dihydrogen ($H_{2}$) is the most stable and commonly found form of hydrogen, which is widely applied in industrial processes and as a clean energy source.

Building upon this framework, the study adopts a systematic labeling approach to clearly and accurately identify and classify VOCs. For instance, the label **10_7:$H_{2}$: Dihydrogen** is used to classify dihydrogen, where
**10_7** represents the component category;$H_{2}$ specifies the type of component;and **Dihydrogen** corresponds to the standard chemical name.

To provide additional clarity on the labeling process, refer to the example illustrated in Figure 4; 4_14: $H2TC_{9}H_{20_{M}}$ corresponds to the compound *hexane, 2,2,4-trimethyl* and is structured so that each part provides specific information about the component. The prefix **4_** identifies the general category of the component, while **14** indicates its specific type within that category. The section **H2T** acts as a key identifier to clarify that the compound is not *heptane, 2,4-dimethyl* (another isomer with the same molecular formula) but *hexane, 2,2,4-trimethyl*, highlighting its unique branched structure. The formula **C_9_H_20_** represents the chemical composition of the compound, with nine carbon atoms and 20 hydrogen atoms. Finally, **M** indicates that this compound has been identified in technical (T) and/or medical (M) studies. This coding system ensures precise description and avoids confusion between isomers sharing the same molecular formula. By employing this systematic approach, the study introduces a structured and precise nomenclature that simplifies the organization, interpretation, and analysis of the VOC-related data. This framework not only enhances the clarity of the study but also helps the reader better understand the diversity and detailed distribution of the chemical components.

The study has identified 107 common VOCs in both medical and technical research. Additionally, 361 exclusive components have been identified solely in medical studies. In technical studies, 7 urine biomarkers and 53 breath biomarkers were examined, while medical studies analyzed 17 urine biomarkers, 1 saliva biomarker, 5 blood biomarkers, and 37 breath biomarkers. The identified VOCs are presented in Figure 5. These figures highlight findings across various cancer types with labeled components (item and label) encompassing all elements identified in the research. These VOCs are distributed among nine cancer types and have been meticulously organized in Table A1.

In the analysis of lung cancer, a total of 59 breath biomarkers and one urine biomarker are used. The results reveal a chemical composition that can be categorized into 228 components, which include 25 acids, 28 aldehydes, 37 alkanes, 2 alkylbenzenes, 14 ketones, 5 aromatic compounds, 22 hydrocarbons, 47 organic compounds, 17 alcohols, 3 esters, 2 benzenes, 1 nitrile, 1 styrene, 4 cysteines, 2 sulfur compounds, 6 alkenes, 11 colorless gases, and 1 chemical [1,4,5,6,7,8,9,10,11,12,13,14,15,18,19,21,29,38,39,49,50,52,53,54,56,60,61,64,65,66,67,68,69,73,74,75,76,79,83,85,86,87,88,89,92,93,96,97,98,101,106,107,114,116,120]. When diagnosing colorectal cancer, a combination of three urine biomarkers, four blood biomarkers, and seven breath biomarkers are used for analysis. These biomarkers help reveal a chemical composition that consists of various categories, including 38 organic compounds, 2 colorless gases, 15 hydrocarbons, 17 alcohols, 10 aldehydes, 25 alkanes, 12 acids, 11 ketones, 7 aromatic compounds, 1 benzene, 2 alkenes, 7 amino acids, 1 alkylbenzene, 2 esters, 1 sulfur compound, and 1 chemical compound.

In total, 152 components are analyzed to detect colorectal cancer. It is important to note that these elements underwent analysis using six medical devices and 24 specialized sensors [1,4,5,6,7,8,9,10,11,12,13,14,15,16,18,19,20,21,23,25,28,29,30,32,34,37,38,39,40,41,42,45,49,50,51,52,54,56,58,60,61,62,63,65,66,68,69,72,73,74,75,76,77,78,79,80,82,83,84,85,86,87,88,89,90,91,92,93,96,97,98,101,103,104,105,106,107,114,116,117,120,121,122].

Bladder cancer can be diagnosed by analyzing one breath biomarker and four urine biomarkers. The chemical composition of these biomarkers includes various categories, such as 1 ester, 10 hydrocarbons, 9 ketones, 17 organic compounds, 3 acids, 8 alcohols, 4 aldehydes, 6 alkanes, 2 styrenes, 1 terpene, 1 aromatic compound, and 1 sulfur compound, totaling 63 components [1,4,5,6,7,8,9,10,11,12,14,17,18,19,20,21,23,25,28,29,37,38,39,45,50,51,54,56,58,61,62,63,65,67,69,72,73,74,75,76,77,78,79,83,85,86,89,92,93,96,97,101,103,105,106,107,115,116,117,118,119,120]. Cervical cancer can be analyzed using two biomarkers found in urine and one biomarker found in blood. Its chemical composition includes 4 acids, 6 alcohols, 18 alkanes, 5 ketones, 1 colorless gas, 4 terpenes, 19 organic compounds, 5 hydrocarbons, 1 aldehyde, and 1 alkylbenzene, totaling to 64 components [5,6,10,18,19,25,39,56,62,65,67,68,73,75,77,92,106,116,120,122]. Prostate cancer can be detected by analyzing three breath biomarkers and seven urine biomarkers. The analysis reveals a chemical composition that is characterized by categories such as 6 aromatic compounds, 3 colorless gases, 2 chemical compounds, 31 organic compounds, 1 hydrocarbon, 3 acids, 4 alcohols, 14 aldehydes, 4 alkanes, 11 ketones, 1 terpene, 2 esters, and 2 nitriles, adding up to a total of 84 components [1,4,5,6,7,8,9,10,11,12,14,15,16,18,19,20,21,22,23,25,28,29,30,34,37,38,39,40,41,42,45,49,50,51,52,54,56,58,59,60,61,62,63,65,66,67,69,72,73,74,75,76,77,78,79,82,83,84,85,86,88,89,90,92,93,96,98,101,103,104,105,106,107,108,109,110,114,116,117,120]. Breast cancer can be analyzed using four breath and four urine biomarkers. This analysis can reveal a chemical composition that includes various categories such as 5 acids, 4 esters, 7 aldehydes, 2 aromatic compounds, 8 ketones, 17 organic compounds, 5 hydrocarbons, 4 alcohols, 6 alkanes, 3 alkylbenzenes, 3 benzenes, 1 styrene, 7 terpenes, and 1 sulfur compound. In total, 73 components can be analyzed for the detection of breast cancer [1,4,5,6,7,8,9,10,11,12,13,14,15,18,19,21,29,38,39,49,50,52,53,54,56,60,61,64,65,66,67,68,69,73,74,75,76,79,83,84,85,86,87,88,89,92,93,96,97,98,101,106,107,114,116,120]. Furthermore, ten breath biomarkers are used for the analysis of gastric cancer, unveiling a chemical composition that includes 4 acids, 5 alcohols, 3 aldehydes, 6 aromatic compounds, 3 esters, 6 hydrocarbons, 20 organic compounds, 7 alkanes, 2 alkenes, 4 ketones, 2 nitriles, 2 sulfur compounds, 2 benzene, and 1 amino acid, totaling 67 components [4,5,6,7,8,9,10,11,12,13,14,15,16,18,19,20,21,22,23,25,28,29,30,32,34,37,38,39,40,41,42,45,49,50,51,52,54,56,58,60,61,62,63,65,68,69,72,73,74,75,76,77,78,79,80,82,83,85,86,87,88,89,90,91,92,93,96,97,98,101,103,105,106,107,110,116,117,120,122]. In the analysis of oral cancer, 1 saliva biomarker is used, comprising a chemical composition that includes five acids, one alcohol, one aldehyde, and two organic compounds (totaling nine components) [4,14,25,75,83,92,106,114]. The study generated a list of significant components, emphasizing those occurring between 10 and 36 occurrences.

#### 4.2.1. VOC-Based Biomarkers for Early Cancer Detection Using eNose Technology

Studies on eNose technology have shown significant promise in cancer detection, primarily through the analysis of blood, breath, salivary, and urine. These methods rely on metrics such as Area under the Curve (AUC), SE, and SP, which is driven by the identification of specific VOCs that reflect the patient’s metabolic state.
**Blood Sample Analysis Results**The analysis of blood samples for *colorectal cancer* detection yields highly promising results, with an AUC of 0.85 (95% CI: 0.737–0.953%), demonstrating robust diagnostic capability. This is further supported by an SE of 83% and SP of 82%, emphasizing its potential for early diagnosis and disease differentiation. Among the identified biomarkers, p-Cresol stands out, achieving an AUC of 0.85 with enhanced metrics of SE = 83.3% and SP = 84.7%, as corroborated by prior studies [82,87]. Chemical analysis highlights several VOCs associated with the metabolic profile of colorectal cancer, including 17_1:$C_{7}H_{8}O$ (retention time: 7.92 ± 0.08 min), 17_2:$C_{8}H_{7}N$ (9.84 min), 10_1:$DLC_{2}H_{6}O$ (8.54 min), and 13_1:$C_{6}H_{6}$ (6.32 min) [82]. These compounds exhibit precise retention times in chromatographic analysis, facilitating accurate detection and characterization in patients with this pathology. For *cervical cancer*, blood sample analysis demonstrates remarkable diagnostic accuracy, with an SE of 89% and SP of 93%, which is particularly effective in early-stage detection. These metrics underscore the efficacy of the biomarkers in distinguishing affected patients from healthy controls, minimizing false negatives and false positives [121]. The blood chemical profile of cervical cancer patients includes several key biomarkers, expressed in international units (iu), such as 4_15:$2PTC_{8}H_{18}=992\;\mathrm{iu}$, 13_4:$2DTC_{16}H_{34}=1320\;\mathrm{iu}$, 4_12:$O2TC_{11}H_{24}=1029\;\mathrm{iu}$, and 4_13:$2UDC_{13}H_{28}=1828\;\mathrm{iu}$. Additional compounds include 1_7:$2MFC_{7}H_{14}O_{2}=917\;\mathrm{iu}$, 4_14:$H2TC_{9}H_{20}=767\;\mathrm{iu}$, and 13_7:$3D1TC_{16}H_{34}=1443\;\mathrm{iu}$, among others [121]. These biomarkers, identified using advanced chromatography and spectrometry techniques, represent potential metabolic markers of cervical cancer, providing high specificity and reliability for diagnostic applications.**Urine Sample Analysis Results**The diagnosis of cancer through biomarkers in urine samples has proven to be a promising and non-invasive tool supported by robust metrics and detailed chemical analyses. For *prostate cancer*, a high AUC (0.92) and significant *p*-values are reported for biomarkers such as 16_4:$C_{18}H_{20}O_{2}$ and 17_13:$C_{9}H_{26}O_{2}Si_{3}$[104], while another study shows slightly lower performance with AUC = 83% (95% CI: 77–89%), SE = 82% (95% CI: 73–88%), and SP = 87% (95% CI: 75–94%) [112]. In the case of bladder cancer, the metrics are outstanding, with an AUC of 96%, SE of 93.3%, and SP of 86.7% [116,119], supported by biomarkers such as 17_64:$1PC_{8}H_{10}O$ ($2.70\times 10^{-3}$), 20_4:$2MPC_{10}H_{18}O$ ($7.0\times 10^{-4}$), and 14_18:$C_{6}H_{5}COCH_{3}$ ($7.0\times 10^{-4}$). For *breast cancer*, the results are equally promising, with an AUC of 0.98 (95% CI: 0.85–100%), SE of 97%, and SP of 72% [65,66], highlighting biomarkers such as 2_13:$2PC_{3}H_{8}O$ (SE = 93.8%, SP = 84.6%) and 17_78:$C_{14}H_{12}N_{2}O$ (SE = 93.8%, SP = 84.6%). In *cervical cancer*, the variability explained by PCA reaches 98.4%, with an SE of 91.6% (61.5–99.7%) and SP of 100% (73.5–100%) [122], and relevant biomarkers such as 1_1:$CH_{3}COOH$ (24.6%), 2_4:$C_{3}H_{8}O$ (28.4%), and 4_6:$C_{16}H_{34}$ (30.9%), positioning this approach as highly reliable. For *colorectal cancer*, although the metrics are somewhat lower (AUC: 0.82, SE: 87.9%, SP: 84.6% [84]), biomarkers such as 2_13:$2PC_{3}H_{8}O$ (AUC = 0.82) and 1_10:$C_{11}H_{22}O_{2}$ (AUC = 0.82) are promising for early detection. In the case of *lung cancer*, although no detailed global metrics are reported, biomarkers such as 14_24:$2PC_{5}H_{10}O$ show perfect performance with SE, SP, and PPV at 100% [10], highlighting their diagnostic utility. Finally, for *pancreatic cancer*, an AUC of 0.92 is reported with an SE of 84% and SP of 94% for PDAC [1,103], although specific chemical compositions are not detailed. These results reflect the potential of urinary biomarkers as a high-impact diagnostic tool, although variability in some metrics underscores the need for greater standardization and research.**Breath Sample Analysis Results**Breath analysis has demonstrated significant potential as a non-invasive diagnostic tool for various cancers, leveraging the VOCs associated with disease-specific metabolic alterations. To begin with, for *breast cancer*, an AUC of 86% indicates strong discriminatory power, with a Negative Predictive Value (NPV) exceeding 99.9% and a Positive Predictive Value (PPV) of 100%, ensuring minimal false negatives and the perfect identification of positive cases [67]. Notably, key VOCs include 4_24:$NC_{9}H_{20}$, 4_4:$C_{13}H_{28}$, 13_25:$5MUC_{12}H_{26}$, and 4_44:$3MPC_{16}H_{34}$, which highlight disruptions in lipid peroxidation pathways characteristic of tumor cell activity. In addition, for *colorectal cancer*, an AUC of 91% underscores excellent diagnostic performance, with an SE of 85% and SP of 94%, effectively differentiating between healthy and diseased individuals [88]. For instance, notable biomarkers include 2_13:$2PC_{3}H_{8}O$ (SE: 71.4%, SP: 90.9%), $3\_8:C_{2}H_{4}O$ (SE: 70.2%, SP: 90.7%), and $1\_8:C_{3}H_{6}O$ (SE: 72.1%, SP: 89.9%), which reflect oxidative stress and sulfur metabolism. Moreover, for *gastric cancer*, breath analysis achieves an AUC of 97%, with an SE of 100% and SP of 93%, demonstrating exceptional accuracy [74]. Specifically, biomarkers such as 5_1:$C_{12}H_{24}$, 4_3:$C_{18}H_{38}$, 8_11:$MC_{8}H_{10}$, and 4_6:$C_{16}H_{34}$ correlate with metabolic disruptions associated with the disease, offering precise detection capabilities. Similarly, for *lung cancer*, the diagnostic method achieves an AUC of 92%, SE of 96%, and SP of 88%, demonstrating robust performance [8]. In this context, key VOCs include $17\_35:C_{5}H_{8}$, 1_8:$C_{3}H_{6}O$, 4_31:$C_{5}H_{12}$, and 8_4:$C_{6}H_{5}CH_{3}$, which reflect alterations in volatile profiles due to tumor activity. Furthermore, for *pancreatic cancer*, an AUC of 0.92, SE of 87.67%, and SP of 87% highlight its effectiveness in early detection [97]. Notably, VOCs such as 3_23:$CH_{2}O$, 17_62:$AC_{4}H_{8}O_{2}$, 4_34:$C_{11}H_{24}$, and 4_33:$C_{4}H_{10}S$ provide a unique metabolic profile for pancreatic cancer identification. Lastly, for *prostate cancer*, the diagnostic accuracy is underscored by an AUC of 95%, with VOCs such as 1_34:$AHC_{4}H_{6}O_{3}$, 12_8:$PYC_{6}H_{12}O_{2}$, and 14_39:$EVC_{5}H_{8}O$ serving as potential biomarkers [109]. On the other hand, another study reported slightly lower metrics (AUC 0.75, SE and SP at 84%), but the findings still demonstrate robust diagnostic capability [110].**Analysis of results in Salivary samples**Salivary analysis demonstrates exceptional efficacy as a diagnostic tool for *oral cancer*, with an AUC of 90%, reflecting an excellent ability to distinguish between patients with and without the disease. The method achieves an SE and SP of 100%, ensuring the accurate identification of all positive cases and the absence of false positives, making it a highly reliable diagnostic approach. Key VOCs identified in saliva include 2_9:1$OC_{8}H_{16}O=1.36\times 10^{-4}$, 1_4:$C_{6}H_{12}O_{2}=4.93\times 10^{-2}$, 3_24:$C_{8}H_{14}O=4.74\times 10^{-2}$, 1_5:$C_{7}H_{14}O_{2}=1.06\times 10^{-2}$, 1_15:$C_{8}H_{16}O_{2}=1.06\times 10^{-2}$, 17_56:$C_{9}H_{16}O=1.36\times 10^{-4}$, 17_63:$C_{9}H_{18}O_{2}=4.74\times 10^{-2}$, 3_25:2$DC_{10}H_{16}O=2.07\times 10^{-2}$, and 1_16:$C_{11}H_{20}O_{2}=1.06\times 10^{-2}$. Among these, the high concentrations of $C_{6}H_{12}O_{2}$ and $C_{11}H_{20}O_{2}$ ($4.93\times 10^{-2}$) stand out for their relevance in the pathological processes of oral cancer. These compounds reflect specific metabolic alterations associated with the disease, enabling the development of precise metabolic profiles essential for accurate and non-invasive diagnosis [114].

#### 4.2.2. Metabolic Patterns and Potential Biomarkers in the Identification of VOCs Associated with Cancer

The analysis of the study revealed a total of 661 chemical components. Figure A1, Figure A2, Figure A3, Figure A4, Figure A5, Figure A6 and Figure A7 provide a comprehensive visualization of these components, which are organized into 20 specific categories. These graphical representations offer an overview of the diversity and distribution of the components identified in the study as well as a detailed explanation of their respective labeling.

The identification of VOCs across multiple types of cancer represents a significant advancement in the development of non-invasive tools for diagnosis and disease monitoring. These findings highlight diverse metabolic patterns with both diagnostic and therapeutic implications, positioning VOCs as key indicators in cancer characterization. In this study, particular emphasis is placed on the identification of VOCs and their distribution among various cancer types. These compounds exhibit distinct properties, including molecular characteristics and their potential to serve as biomarkers for early diagnosis, disease progression tracking, and the development of personalized therapeutic strategies.

Out of a total of 661 compounds detected, 77.09% (508 compounds) were unique, highlighting the metabolic diversity and biochemical heterogeneity across various types of cancer. This prevalence of unique compounds underscores the need for comprehensive analytical approaches that address both less frequent and recurring compounds. Additionally, 105 compounds (15.91%) were identified with recurrence rates of two to five detections, which were associated with less common metabolic patterns or specific cancer subtypes. This group may provide valuable insights for differentiated diagnoses in atypical metabolic contexts. Furthermore, compounds detected six to nine times (48 compounds, 7.26%) form an intermediate group that could reflect specific metabolic processes in cancers with moderate incidence or at particular stages of progression.

Figure 6 shows the frequency of VOC identification in cancer studies. Among these compounds, those associated with critical metabolic alterations stand out. Detailed information on all 120 studies, including compacted and relevant data, can be found in Appendix C. The compounds, those associated with critical metabolic alterations, such as *acetone* ($C_{3}H_{6}O$), *benzene* ($C_{6}H_{6}$), and *toluene* ($C_{6}H_{5}CH_{3}$), were detected 36 (5.45%), 34 (5.14%), and 27 (4.08%) times, respectively. These recurrent VOCs are of interest as potential universal biomarkers due to their association with common metabolic processes, such as oxidative stress and disruptions in energy metabolism. Furthermore, compounds such as *ethanol* ($C_{2}H_{6}O$) and *hexanal* ($C_{6}H_{12}O$), detected 24 (3.63%) and 20 (3.03%) times, respectively, may reflect specific metabolic disruptions with more specialized diagnostic applications. In an intermediate recurrence range, compounds like *ammonia* ($NH_{3}$) with 15 detections (2.27%) and *methane* ($CH_{6}$) with 14 detections (2.12%) suggest their potential involvement in relevant metabolic processes that, while not exclusive to cancer, could indicate significant biological alterations. Finally, less frequent compounds, such as *pentanal* ($C_{5}H_{10}O$), *styrene* ($C_{8}H_{8}$), and *butanal* ($C_{4}H_{8}O$), each detected 10 times (1.51%), are linked to specific metabolic contexts or rarer cancers. Despite their low recurrence, these VOCs could be critical in differential diagnoses and the early detection of cancer subtypes or initial stages of the disease.

The following section will delve into the most relevant biomarkers, their relationship with altered metabolic processes in cancer, and their potential impact on clinical practice.

*Ethylbenzene* (9_1:$ETC_{8}H_{10_{TM}}$) is an organic compound with a benzene ring bonded to an ethyl group ($-C_{2}H_{5}$), which is notable for its relevance in cancer detection. Its presence has been documented in lung, colorectal, and gastric cancers, highlighting its potential as a biomarker. Research [8,68,86] has extensively analyzed its specificity and sensitivity, underscoring its diagnostic utility. Further studies [25,92,93] explored its retention time during chromatographic analysis, offering valuable insights into its behavior in detection methodologies. Additionally, its concentration in pancreatic cancer was examined in [78], while the statistical validation of its presence in cancer-related samples was reinforced by *p*-value assessments in [54]. Complementary investigations [95] have further evaluated its diagnostic performance, reinforcing its significance in medical research. Structurally, ethylbenzene comprises eight carbon atoms and ten hydrogen atoms, as denoted by its molecular formula, $C_{8}H_{10}$.

*Hydrogen* (16_1:$H_{T}$), the simplest and most reactive element, has emerged as a noteworthy component in cancer diagnostics. Its role in detecting lung and colorectal cancers has been demonstrated across several studies [23,28,30,40,42], which evaluated its specificity and sensitivity. Furthermore, investigations into its retention time during analysis [41,45,51,103] provided critical data on its analytical properties. Advanced sensor technologies were employed in [63] to assess hydrogen’s sensitivity, confirming its applicability in distinguishing cancer types. Composed of two hydrogen atoms bonded together, hydrogen forms the diatomic molecule $H_{2}$, whose simplicity belies its diagnostic potential in modern oncological research.

*Ammonia* (10_12:${NH}_{3_{TM}}$), a pungent gaseous compound integral to the natural nitrogen cycle [136], has also shown promise as a cancer diagnostic marker. Its detection in lung and colorectal cancers has been supported by studies [23,28,30,34,88], which analyzed its specificity and sensitivity. The compound’s concentration in various contexts, including cancer, was further studied in [101], while its role in metabolic profiling through odor density evaluation was detailed in [40]. Its retention time during chromatographic analysis was rigorously investigated in [41,42,45,51,103], solidifying its role in diagnostic processes. Moreover, studies [56,60] have specifically linked ammonia concentrations to lung cancer, and sensor-based sensitivity evaluations [63] have demonstrated its efficacy in diverse detection systems. Structurally, ammonia consists of one nitrogen atom and three hydrogen atoms, forming the molecule $NH_{3}$, whose diagnostic implications are increasingly recognized in the field of oncology.

*Toluene* (8_4:$C_{6}H_{5}CH_{3_{TM}}$) is a versatile chemical compound widely employed in the synthesis of benzoic acid, phenol, and methyl benzoate [136]. Its detection in cancers such as lung, gastric, and prostate cancers has emphasized its significance as a potential biomarker. Various studies [4,6,19,21,23,28,29,38,39,56,58,61,78,101,107] have explored its diagnostic relevance, focusing on aspects like intensity, concentration, specificity, and sensitivity [21,23,28,29,78]. Statistical validations using *p*-values have further substantiated its presence in cancer-related samples [19,107]. Insights into its metabolic role have been gained from analyses of its retention time during VOC testing [38,58], while its sensitivity across various sensor technologies has been confirmed [61]. Structurally, toluene features a benzene ring ($C_{6}H_{5}$) bonded to a methyl group ($-CH_{3}$), as represented by its molecular formula.

*Acetone* (1_8:$C_{3}H_{6}O_{TM}$) is a naturally occurring metabolite known for its solvent properties and rapid evaporation. Its detection in cancers, including lung, gastric, pancreatic, and colorectal cancers, underscores its diagnostic potential. Extensive studies [5,8,10,68,85,86,87,88,97,98] have evaluated its specificity and sensitivity, while its intensity in cancer samples was investigated in [72]. Statistical confirmation of its presence using *p*-values has been detailed in [96]. Concentration analyses in gastric and lung cancers have been conducted [16,19] alongside studies of its retention time [20]. Comprehensive evaluations of its specificity and sensitivity have further supported its role as a biomarker [23,28,30,34,38]. Additional research has investigated its odor density significance [40] and its performance in sensor systems [62,63]. Structurally, acetone is composed of three carbon atoms, six hydrogen atoms, and one oxygen atom, forming the molecule $C_{3}H_{6}O$.

*Ethanol* (2_24:$C_{2}H_{6}O_{TM}$), a widely used alcohol in chemical and pharmaceutical applications [136], has also been identified as a biomarker in colorectal and lung cancers. Research [23,28,30,34,38,85,86,87,88] has highlighted its specificity and sensitivity in cancer diagnostics. Its retention time during analysis has been investigated in [37,41,42,51,92,103], while its concentration in lung cancer was specifically examined in [56,60,61,101]. Further studies have explored ethanol’s role in odor density profiling [40] and its performance across different sensor technologies [62,63]. Structurally, ethanol consists of two carbon atoms, six hydrogen atoms, and one oxygen atom, which are encapsulated in the molecular formula $C_{2}H_{6}O$.

*Nonanal* (3_1:$C_{9}H_{18}O_{TM}$) is a noteworthy aldehyde widely used in analytical and sensory studies [136]. Its detection in a broad range of cancers, including pancreatic, lung, gastric, bladder, colorectal, prostate, and breast cancers, highlights its diagnostic relevance across multiple oncological contexts. Extensive research [1,92,93,116] has examined its retention time, while studies [4,6,107] employed *p*-values to validate its presence in cancer-related samples. Nonanal’s specificity and sensitivity have been rigorously evaluated in investigations such as [5,69,85,106], emphasizing its potential as a biomarker. Furthermore, research into its concentration in lung cancer [11,12,19] has provided deeper insights into its metabolic implications. Structurally, Nonanal comprises nine carbon atoms, eighteen hydrogen atoms, and one oxygen atom, reflecting its molecular formula $C_{9}H_{18}O$.

*Hexanal* (3_2:$C_{6}H_{12}O_{TM}$), another aldehyde, originates from biochemical processes in plants [136]. It has been identified in lung, colorectal, gastric, and breast cancers, demonstrating its potential as a diagnostic marker. P-value-based studies [4,6,69] have confirmed its presence, and its role in lung cancer detection precision has been explored in [7]. Comprehensive analyses of its specificity and sensitivity are documented in [8,10,15,29,86], while research on its concentration in lung cancer [12,13,39] underscores its metabolic significance. Investigations into its retention time [76,92,93] further support its diagnostic utility. Hexanal’s ability to distinguish cancer patients from healthy controls is detailed in [19], and its sensitivity in sensor technologies has been analyzed in [54,56,61,63]. Structurally, Hexanal consists of six carbon atoms, twelve hydrogen atoms, and one oxygen atom, as denoted by $C_{6}H_{12}O$.

*Octanal* (3_7:$OTC_{8}H_{16}O_{TM}$) is an organic compound in the aldehyde family, which is known for its role in contributing to the aroma and flavor of various foods and products [136]. It has been detected in cancers such as lung, prostate, and colorectal cancers, underscoring its potential as a biomarker. Studies [5,106] have evaluated the specificity and sensitivity of octanal in cancer detection, while its presence in cancer-related samples was statistically validated through *p*-values in [6,69]. The concentration of octanal in lung cancer has been extensively examined in [11,12,39] with additional research [58,92,93] investigating its retention time during VOC analyses. Furthermore, its sensitivity to lung cancer detection was assessed in [54]. Structurally, octanal is composed of eight carbon atoms, sixteen hydrogen atoms, and one oxygen atom, as represented by its molecular formula $C_{8}H_{16}O$.

*Benzene* (13_1:$C_{6}H_{6_{TM}}$), a stable aromatic compound, is widely utilized in industrial applications, including the synthesis of resins, synthetic fibers, pharmaceuticals, and dyes [136]. Its detection in colorectal, lung, pancreatic, and gastric cancers [20,37,41,42,51,74,82,93,103] underscores its significance as a biomarker. Studies on benzene’s retention time [20,37,41,42,51,74,82,93,103] have offered critical insights into its behavior during analytical testing. Its specificity and sensitivity have been extensively evaluated [5,8,18,23,28,29,30,34,38,56,61,80,88,98], with research [7] employing benzene to improve the accuracy of lung cancer detection. Additional studies [16,21,39,60,78,101] focused on its concentration in gastric and lung cancers, while its role in differentiating cancer patients from healthy individuals has been highlighted in [32]. Benzene’s odor density [40] and sensitivity in sensor applications [63] further validate its relevance in diagnostic methodologies. Structurally, benzene features an aromatic ring composed of six carbon and six hydrogen atoms, which is represented by $C_{6}H_{6}$.

*Heptanal* (3_10:$HPC_{7}H_{14}O_{TM}$) is a versatile aldehyde widely used in the fragrance and flavoring industries [136]. Its detection in cancers such as lung, prostate, breast, colorectal, and pancreatic cancers underscores its diagnostic potential. Studies [6,69,107] have employed *p*-values to confirm its presence in cancer-related samples, while its specificity and sensitivity have been extensively analyzed in [10,15,29,56,86,106]. Additionally, heptanal has proven effective in distinguishing individuals with breast and lung cancers from healthy controls [66]. Concentration-specific studies focusing on lung cancer [11,12] and investigations into its retention time during VOC separation processes [58,93] have further highlighted its metabolic and diagnostic significance. Structurally, heptanal comprises seven carbon atoms, fourteen hydrogen atoms, and one oxygen atom, as represented by $C_{7}H_{14}O$.

*Decane* (4_25:$C_{10}H_{22_{TM}}$), a saturated hydrocarbon from the alkane family, is characterized by its stable single bonds between carbon and hydrogen atoms [136]. It has been identified in pancreatic, lung, colorectal, and breast cancers, indicating its relevance as a biomarker in cancer detection. Research [5,10,29,38,56,61] has evaluated its specificity and sensitivity, providing robust evidence of its diagnostic utility. Its concentration in lung cancer has been studied in [13,16,60], while its retention time during VOC separation processes was analyzed in [21,92]. The molecular structure of decane, with ten carbon atoms and twenty-two hydrogen atoms, highlights its simplicity and stability, making it an essential target for biochemical analyses.

*Isobutane* (4_72:$C_{4}H_{10_{T}}$), an aliphatic hydrocarbon with a low boiling point, is frequently employed in applications requiring volatile compounds [136]. Its detection in the lungs of cancer patients has demonstrated its potential as a diagnostic biomarker. Studies [23,28,29,30,34] have investigated its specificity and sensitivity, while its concentration in lung cancer was examined in [39]. Isobutane’s role in odor density profiling has been explored in [40], and its residence time during VOC separation processes has been detailed in [41,42,45,51]. Furthermore, its sensitivity across diverse sensor technologies was evaluated in [63]. Structurally, isobutane consists of four carbon atoms and ten hydrogen atoms, forming the compact and reactive molecule $C_{4}H_{10}$.

*Methane* (4_65:$CH_{4_{T}}$), a major component of natural gas, is primarily derived from fossil fuel extraction and microbial activities [136]. Its detection in lung cancer underscores its potential diagnostic relevance. Studies [28,30,34,59,80] have analyzed methane’s specificity and sensitivity in cancer detection, while its retention time during testing has been explored in [37]. Research into its concentration in lung cancer was conducted in [101], and its role in determining odor density was highlighted in [40]. Furthermore, methane’s retention time during VOC separation processes has been investigated in [41,42,45,51,103], and its sensitivity across various sensor technologies was evaluated in [63]. Structurally, methane consists of one carbon atom bonded to four hydrogen atoms, forming the simplest hydrocarbon, $CH_{4}$.

*Styrene* (13_39:$C_{8}H_{8_{TM}}$) is a monomer extensively utilized in the production of polystyrene and other polymeric materials [136]. Its detection in lung and stomach cancers highlights its potential as a biomarker. Studies [5,10,29] have analyzed its specificity and sensitivity in cancer detection, while research [7] demonstrated its role in improving lung cancer diagnostic accuracy. Further investigations [19,39,60,78,101] examined styrene’s concentration in stomach and lung cancers, and its sensitivity in lung cancer detection was assessed in [54]. Structurally, styrene comprises eight carbon atoms and eight hydrogen atoms, forming the molecular formula $C_{8}H_{8}$.

*Cyclohexane* (17_12:$C_{6}H_{10}O_{M}$), commonly used as a solvent in industrial chemical production, has been associated with cancers such as cervical, lung, prostate, bladder, stomach, and colorectal cancers [136]. Its specificity and sensitivity were thoroughly evaluated in [10,18,106,120], while *p*-values confirmed its presence in cancer-related samples [6,14,117]. Studies [76,89,116] investigated its retention time during VOC analyses, and its concentration in gastric and lung cancers was specifically examined in [11,19]. Cyclohexane consists of six carbon atoms, ten hydrogen atoms, and one oxygen atom, as denoted by its molecular formula $C_{6}H_{10}O$.

*Pentanal* (17_33:$PTC_{5}H_{10}O_{TM}$) is an organic aldehyde often employed in the fragrance industry for its distinct aromatic characteristics. Its detection in lung and prostate cancers has established its diagnostic potential. Studies [5,52,106] explored its specificity and sensitivity, while its presence was statistically validated in [6]. Research [7] highlighted its role in lung cancer detection accuracy, and its concentration in lung cancer was detailed in [11,12,39]. Additionally, its retention time during VOC analyses was analyzed in [20], and its sensitivity in lung cancer detection was evaluated in [56]. Pentanal’s molecular structure includes five carbon atoms, ten hydrogen atoms, and one oxygen atom, forming $C_{5}H_{10}O$.

*Isoprene* (17_35:$C_{5}H_{8_{TM}}$), classified as a diene due to its two conjugated double bonds, is a hydrocarbon crucial for metabolic profiling [136]. Its presence in lung, stomach, colorectal, and prostate cancers highlights its biomarker significance. P-values have validated isoprene’s detection in cancer-related samples [6,77], while its intensity was analyzed in [72]. Studies [8,52,88,110] examined its specificity and sensitivity, and its retention time during VOC analyses was studied in [20,74]. Research into isoprene’s concentration in lung cancer was conducted in [19,39,60], while its diagnostic sensitivity was further evaluated in [54,56]. Its performance in sensor technologies was tested in [62]. Structurally, isoprene consists of five carbon atoms and eight hydrogen atoms, which are represented by $C_{5}H_{8}$.

*Butanal* (17_43:$BTC_{4}H_{8}O_{TM}$) is an organic aldehyde formed during the oxidation of butyl alcohols, and it is recognized for its potential as a diagnostic biomarker in cancers such as lung, prostate, and gastric cancers. Its role in evaluating the accuracy of lung cancer detection was demonstrated in [7], and its concentration in lung cancer was further analyzed in [12,16,39]. Additionally, the retention time of butanal during VOC analyses was explored in [20], while its sensitivity in lung cancer detection was assessed in [54,56]. Structurally, butanal is composed of four carbon atoms, eight hydrogen atoms, and one oxygen atom, and it is represented by $C_{4}H_{8}O$.

*Phenol* (17_134:$C_{6}H_{6}O_{TM}$), an organic compound with acidic properties due to its hydroxyl group, has been detected in a variety of cancers, including cervical, prostate, colorectal, gastric, bladder, and lung cancers [136]. Studies [75,86,120] have analyzed the specificity and sensitivity of phenol, while its retention time and intensity during VOC analyses were examined in [25,83,105,116]. P-values were employed in [72] to validate its presence in cancer-related samples, further substantiating its diagnostic potential. Structurally, phenol consists of six carbon atoms, six hydrogen atoms, and one oxygen atom, as denoted by $C_{6}H_{6}O$.

*Benzaldehyde* (17_49:$C_{7}H_{6}O_{TM}$) is a versatile compound commonly utilized in industrial applications and organic synthesis with significant detection in cancers such as lung, breast, colorectal, bladder, pancreatic, and prostate cancers [136]. Research [18,50,65,97,106] has examined its specificity and sensitivity, while its retention time during VOC analyses was explored in [92,116]. Investigations into its concentration in lung cancer were conducted in [11,14], and its presence was validated through *p*-values in [69,117]. Moreover, its sensitivity in lung cancer detection was further evaluated in [56]. Benzaldehyde’s molecular structure comprises seven carbon atoms, six hydrogen atoms, and one oxygen atom, making it a critical target in metabolic profiling.

### 4.3. Analysis of Techniques

The analysis identified 34 techniques, classified into five categories, as shown in Figure 7. These techniques were frequently used in investigations to evaluate the performance of specific concentrations of components in the reviewed studies. Each technique has been labeled and associated with the types of cancer identified in Table 5. The following section provides a detailed analysis of the most frequently used techniques.

#### 4.3.1. Support Vector Machine (CVM-1)

Support Vector Machine (SVM) is a supervised learning technique used for classification and regression. It is based on finding the optimal hyperplane that separates different classes of data in a multidimensional space. In DM [109], flexible models were developed to maximize sensitivity or specificity. In DT [23], CVM-1 was used to obtain optimal parameters. The technique was utilized for various purposes, including addressing category imbalances [28], minimizing structural risks [29,30,31,33,52,63], achieving accuracy in multiclass classification [34], integrating clinical features [37], reducing dimensions [39,50,59,70,71], performing cross-validation [101,103], and analyzing specific datasets such as sensor gases [40] and lung cancer detection [41]. Additionally, it was used to project features into higher-dimensional spaces [51,54]. Equation (Equation 1) for CVM-1 is as follows: where w represents the slopes of the lines, b is the bias term, $x_{i}$ is a feature vector, $y_{i}$ is the corresponding class label for sample $x_{i}$ (1 or −1 in binary classification), $\zeta_{i}$ are slack variables that represent the allowed classification error for sample *i*, and C is a hyperparameter that controls the trade-off between the width of the margin and the number of errors allowed.
(1)$$\begin{array}{c} min\frac{1}{2}{\left|\left|w\right|\right|}^{2}+C\sum_{i=1}^{n}\zeta_{i}\\ y_{i}\left(w^{t}x_{i}+b\right)\geq 1-\zeta_{i} \end{array}$$

#### 4.3.2. K-Nearest Neighbor (CCL-4)

K-Nearest Neighbor is a supervised machine learning technique based on the principle that similar elements are close to each other. In [23,29,30,45], it was used to determine the number of nearest neighbors and the distance metric. In [70,80], the data were used to calculate four factors: true positive (TP), false positive (FP), true negative (TN), and false negative (FN). In [34], vectors of known classes were used for each sample in the training set. In [39], the classifier performance was comparatively analyzed using the Canberra distance without normalization. In [40], the classifier processed the distinctive features extracted from the data. In [50], the classifier distinguished individuals as either ‘sick’ or ‘healthy’ based on the training set. In [59,63], the CCL-4 technique was applied to determine the classifier accuracy. Equation (Equation 2) is as follows: where (x) ‘point’ refers to the point in the training data.
(2)$$\begin{array}{c} W\left(x\right)=Validalitas\left(x\right)x\frac{1}{d_{e+0.5}} \end{array}$$

#### 4.3.3. Principal Component Analysis (CR-10)

Dimensionality reduction is a technique used to simplify complex datasets while retaining as much information as possible. In the context of MD, a statistical analysis is performed to find correlations between variables and reduce them to a smaller number of principal components [120]. The normalization of each sample was carried out in [83,109] before using the CR-10 technique with the processed data. In [121], the analysis examined all biomarkers to identify clustering trends at a 5% significance level. In [114], it verified the distribution of VOCs based on biomarker groups. Additionally, in TD [25,33,58,101,119,122], the extracted features were used to build a model distinguishing between cancer patients and healthy controls. In some cases, it is used to reduce the set of components and obtain the greatest number of variations in the sensors [50,54,69,99,111]. In other cases, it is used to find the main components that maximize the variation in the sensor responses [32,39,53,71]. In [46,103], a multivariate component was created by merging the variables of interest. The use of CR-10 is prevalent in the literature. Finally, in [59], CR-10 was applied to reduce the dimensionality of the dataset. In [61], the efficiency of discriminating between classes, as well as classifying characteristics and variables, is discussed. Equation (Equation 3) from CR11 is as follows, where N is the normal, and u is the mean of the normal distribution N (u,C), implying a probability distribution over T,S is the covariance matrix.
(3)$$\begin{array}{c} \frac{1}{N}\sum_{n=1}^{N}{\left\{u_{1}^{T}x_{u}-u_{1}^{T}\bar{x}\right\}}^{2}=u_{1}^{T}Su_{1} \end{array}$$

#### 4.3.4. Artificial Neural Network (CR-12)

Machine learning models are inspired by the functioning of the human brain. They are applied in various tasks, such as pattern recognition in images, speech, and natural language processing. In [110], multiple models were created to explain the correlations between vectors. In [27,36,45], they were used for the pattern recognition of VOCs to differentiate patients with benign conditions from those with malignant neoplasms. In [31] was utilized to classify breath biomarkers with a sensitivity of 94% and a specificity of 33%. In [32,52], it was employed to determine the network structure, connection intensity, and processing during training modes. In [81], normalized data were used to train the CR-12. In [95,100], it was used to train the vectors representing the VOCs patterns. At [50], it was utilized to differentiate between lung cancer patients and healthy controls. In [63], it was used to measure the accuracy rate of healthy individuals, which was 88.8%. Equation (Equation 4) of the technique is as follows: where Z is the symbol for denoting the graphical representation of ANN, $W_{\mathrm{is}}$ are the beta coefficients, $X_{\mathrm{is}}$ are the independent variables, and intercept = $W_{0}$.
(4)$$\begin{array}{c} Z=Bias+W_{1}X_{1}+W_{2}X_{2}+\ldots \ldots +W_{n}X_{n} \end{array}$$

#### 4.3.5. Naive Bayes (CP-2)

The technique used for classification tasks involves using characteristics (independent variables) to predict the class or category to which a given data point belongs. This method was used in [23,28,70] to calculate the precision of the components. In [30,34], it was used to calculate the probability of incorporating new data into any of the existing classes using sample data. In [40], it was applied to predict a test from unlabeled data and recognize classes from the input signal. Equation (Equation 5) of the technique is as follows: where P(C|X) is the probability of observing the feature set X given the class C, P(X|C) are the beta coefficients, P(C) is the prior probability of class C, and P(X) is the prior probability of observing the feature set X.
(5)$$\begin{array}{c} P\left(C|X\right)=\frac{P(X|C).P(C)}{P(X)} \end{array}$$

#### 4.3.6. Extreme Gradient Boosting (CA-18)

The Extreme Gradient Boosting machine learning technique is used for classification and regression problems. It involves training a series of models sequentially. In this specific case, ref. [31] was used to discriminate the data and determine a cutoff value for the training set, resulting in high sensitivity and Negative Predictive Value. Additionally, ref. [8] was used to predict the occurrence of lung cancer based on quantitative measurements of VOCs, while in [40], it was applied to predict test data from unlabeled data and recognize input classes. Overall, the use of machine learning techniques has proven to be effective in various applications. Additionally, ref. [8] was used to predict the occurrence of lung cancer based on quantitative measurements of VOCs, while in [40], it was applied to predict test data from unlabeled data and recognize input classes. Finally, ref. [42] also employed this technique. CA-XGBoost provides higher performance by adjusting overlearning with a more regular pattern format. It was used in t107 due to its successful predictions and ease of calculation. Equation (Equation 6) of the technique is as follows: L(t) represents the objective function at iteration t of the training process, n is the number of biomarkers in the dataset, and $l(y_{i},{\hat{y}}_{i})$ is the p-loss function that evaluates the discrepancy between the predicted value (${\hat{y}}_{i}$) and the true label ($y_{i}$) for each sample (i). The loss function for a given problem (e.g., regression, classification) may vary. The variable k represents the number of trees used in the model, and $\Omega (f_{t})$ represents the regularization applied to the model. Regularization (Ω) controls model complexity and helps prevent overfitting by penalizing more complex models. Regularization is applied to each tree ($f_{t}$) added sequentially during the boosting process.
(6)$$\begin{array}{c} L\left(t\right)=\sum_{i=1}^{n}l\left(y_{i},{\hat{y}}_{i}\right)+\sum_{t=1}^{k}\Omega \left(f_{t}\right) \end{array}$$

#### 4.3.7. Random Forest (CA-19)

Random Forest is a supervised learning technique used for classification and regression tasks. It assembles multiple decision trees to make accurate and stable decisions. In [31,40,50], a decision tree ensemble was used to discriminate data and calculate sensitivity and specificity. In [70], it was used to identify molecular subtypes of breast cancer. In [42], it was used to randomly select variables while forming the tree. In reference [52], the tool was utilized to identify potential correlations between model predictors and disease status. Reference [56] employed the tool to develop two diagnostic models based on VOCs peak areas and their ratios. Reference [60] utilized the tool to classify the respiratory patterns of both lung cancer patients and healthy controls.

#### 4.3.8. Partial Least Squares-Discriminant Analysis (CD-3)

CD-3 is a multivariate analysis technique that combines the characteristics of partial least squares (PLS) regression with the discrimination capability of linear discriminant analysis (LDA). It was used in [120] to identify differential sensors between groups and classify the sensor response. In [4], further discrimination between different VOCs was performed. In reference [5], CD-3 was used to discriminate between lung cancer patients and healthy volunteers. In [14], it was examined whether the dataset could be sort by weekly categories. In [119], it was used to obtain the correlation between VOCs. In [122], it was used to classify sensor responses.

#### 4.3.9. Linear Discriminant Analysis (ESC-1)

Linear discriminant analysis is a classification technique used to identify the optimal linear combination of features that distinguish between multiple classes in a given dataset. ESC-1 has been employed in signal processing and independent statistical analyses that rely on biomarkers such as [24,39,40]. In reference [119], the analysis of two groups was performed, correctly identifying all 24/24 (100%) cases of cancer and 70/74 (94.6%) controls. In [30], the authors utilized this method to maximize the distance between classes of VOCs, while [34] used it to calculate the between-class variance and maximize the dataset. In [50], it was used with confidence to evaluate the dimensionality of the data. Additionally, in [55], the method was confidently used to compare direct human breath biomarkers, support materials, fleece masks, and fleece in glass tubes, and confidently classify them into separate groups. Equation (Equation 7) of the technique is as follows: where $u_{1}$ and $u_{2}$ are the datasets.
(7)$$\begin{array}{c} S_{b}=\sum_{j}\left(u_{j}-u_{3}\right)\times {\left(u_{j}-u_{3}\right)}^{T} \end{array}$$

#### 4.3.10. Logistic Regression (RA-1)

Logistic Regression is a statistical technique used for classification and dimensionality reduction. Its goal is to find a linear combination of features that maximizes the separation between multiple classes within a dataset. In [84], it was used to identify potential biomarkers. In [31], it was used to determine the presence of lung cancer, and a cutoff value was applied to calculate the probability. In [34], RA-1 was used for binary classifiers. In [50,60], it was used for estimation of the predictive likelihood of the test data. In [52], it was used to create a diagnostic model with the highest sensitivity and specificity. Equation (Equation 8) of the technique is as follows: where P(y=1|x) represents the conditional probability that the dependent variable (y) equals 1 (or belongs to the positive class) given the feature vector x, e is the natural logarithm basis, $\beta_{0},\beta_{1},\ldots ,\beta_{p}$ are the coefficients (parameters) that are estimated during the training of the model, and $\beta_{0},\beta_{1},\ldots ,\beta_{p}$ are the characteristics or independent variables.
(8)$$\begin{array}{c} P\left(y=1|x\right)=\frac{1}{1+e^{-(\beta_{0}+\beta_{1}x_{1}+\beta_{2}x_{2}+\ldots ++\beta_{p}x_{p})}} \end{array}$$

#### 4.3.11. Cross-Validation (SM-2)

Cross-validation is a statistical technique used to evaluate the performance of a machine learning model and estimate the accuracy of its predictions on unseen data. The results obtained through cross-validation provide a reliable estimate of the model’s performance and can be used to make informed decisions. It has been extensively used in various studies, such as predicting cancer patients from healthy ones in [5,59], validating component discrimination models in [7], and ranking the accuracy of VOCs from highest to lowest in [92] using SM-2. In [55], we used a leave-two-out approach with k equal to the product of the sample sizes of the compared groups. Equation (Equation 9) of SM-2 is as follows: where n is the total number of observations, $y_{i}$ is the response value of the i-th observation, and $f(x_{i})$ is the predicted response value of the *i*-th observation.
(9)$$\begin{array}{c} MSE=\left(\frac{1}{n}\right)*\sum {\left(y_{i}-f\left(x_{i}\right)\right)}^{2} \end{array}$$

#### 4.3.12. K-Fold Cross-Validation (SM-1)

K-fold cross-validation is a machine learning technique that involves dividing the dataset into k approximately equal subsets (folds). In [43], SM-1 was used to enhance the performance of constructing each subclassifier.

#### 4.3.13. Canonical Correlation Analysis (COR-1)

In [21], it was used to demonstrate the levels of VOCs in cancer patients’ breath. Equation (Equation 10) of COR-1 is as follows: where $R_{X}X$ is the covariance matrix of *X*, $R_{Y}Y$ is the covariance matrix of *Y*, $R_{X}Y$ is the cross-covariance matrix between Y and X, and $R_{Y}X$ is the cross-covariance matrix between Y and X.
(10)$$\begin{array}{c} \lambda \left(R_{XX}^{-1}R_{XY}R_{YY}^{-1}R_{YX}\right)\;=\;\left(R_{XX}^{-1}R_{XY}R_{YY}^{-1}R_{YX}\right) \end{array}$$

### 4.4. Sensor Analysis

In technical studies, several sensors were identified. These sensors were grouped into eight categories with a total of 110 devices used to detect various components. You can see the breakdown of these categories in Table A2. Additionally, we provide a detailed description of the most relevant sensors found in this research in Figure 8 and Figure 9.

#### 4.4.1. Gas Sensors

The TGS2610 (SS-30) semiconductor, which utilizes metal oxide technology, is highly effective in detecting signals produced by exhaled gas mixtures. Additionally, it has demonstrated versatility in detecting variations of VOCs in human breath. Its ability to amplify signals and reduce high-frequency noise has been verified through multiple studies [29,30,34,41,42,45,51,60,62]. Similarly, the TGS2602 (SS-1), designed specifically for detecting carbon monoxide (CO), has excelled in numerous applications, including measuring exhaled signals, amplifying signals, reducing high-frequency noise, and classifying responses to signals [23,28,30,34,37,40,41,42,45,51,58,59,60,62,63]. In line with this, the TGS2602 (SS-3), specialized in detecting gases such as nitrogen dioxide ($NO_{2}$) and nitrogen compounds, has demonstrated its effectiveness in measuring signals produced by exhaled gas mixtures and classifying responses to signals [23,28,29,37,45,58,63]. In addition, the TGS826 (SS-4) is a sensor designed for detecting carbon dioxide ($CO_{2}$). It has been tested in various applications, including measuring exhaled signals, detecting variations in VOCs, amplifying signals, reducing high-frequency noise, and classifying responses. Its performance has been evaluated in several studies [23,26,28,29,30,34,41,42,45,51,58,60,62]. On the other hand, the TGS8669 (SS-5) is a sensor specifically designed to detect acetone, and it has demonstrated excellent results in measuring signals generated by the exhaled gas mixture and classifying responses [23,28,58]. Both sensors exhibit distinct characteristics, showcasing the versatility and effectiveness of sensor technology in detecting gaseous components across various applications.

#### 4.4.2. Specialized Sensors

The WSP2110 (SS-12) is a gas concentration detector capable of identifying carbon monoxide (CO), alcohol, and smoke. It has proven effective in the development of systems for detecting changes in human breath, showcasing its versatility [23,26,28,63]. The TGS2600 (SS-25) is designed to detect gases such as nitrogen dioxide ($NO_{2}$), carbon monoxide (CO), and other compounds. This sensor has demonstrated notable versatility in its applications. It can measure exhaled signals, detect variations in VOCs, reduce high-frequency noise, amplify signals, and classify responses [26,28,30,34,37,41,42,45,51,58,59,60,62,63]. The Cyranose 320 (CS-4) is designed to detect and recognize odor and gas patterns. Widely used for creating specific olfactory profiles, it has shown efficacy in recognizing patterns of VOCs, determining component accuracy, and evaluating sensitivity and specificity [33,35,38,39,48,49,50,54,55,70,71,99,102,111,113,119,122]. The Aeonose 320 (CS-4) captures air biomarkers and analyzes patterns of odor and chemical composition. Successfully employed in analyzing biomarkers from exhaled human breath, it has proven its utility in determining component accuracy, evaluating sensitivity and specificity, measuring and analyzing exhaled human breath conductivity, and detecting component recurrence [31,33,35,47,57,69,81,94,95].

#### 4.4.3. Biomarker Capture Sensors

Air sensors play a crucial role in accurately capturing and analyzing various components present in the environment. They have proven to be effective in measuring signals produced by exhaled gases and in analyzing specific concentrations, showcasing their utility not only in medical environments but also in various applications. The MR516 (GS-7) designed to capture air samples and measure gas concentration, has demonstrated its accuracy in measuring signals generated by exhaled gases. Furthermore, it has excelled in analyzing component concentrations in air samples [23,26,28,51,58]. Similarly, the NAP-55A (GS-8), a cost-effective sensor specifically designed for detecting flammable gases, has been assessed in measuring signals produced by gases exhaled from patients. This sensor has proven effective in analyzing VOCs concentrations, highlighting its versatility in environments where cost-effective and efficient detection is essential [26,51,58].

#### 4.4.4. Pattern Analysis and Compositional Analysis Tools Chemistry

Sensors like the W3S (HS-3) and W5S (HS-5) are important for collecting and evaluating air samples. These sensors effectively measure signals generated by exhaled gases from patients by capturing biomarkers and analyzing patterns and chemical composition. As a result, they are useful for the detailed analysis of air composition. Research studies have demonstrated their efficacy in this regard [32,80,93].

#### 4.4.5. Specific Gas Sensors

Sensors like MQ-137 (EC-4), MQ-138 (EC-5), MQ-135 (EC-3), and AS-MLV (PS-7) play a crucial role in monitoring the quality of inhaled air, particularly in medical settings. These sensors are effective and have specific applications for detecting certain gases and pollutants. The MQ-137, for instance, is calibrated to detect ammonia and can measure its concentration in patients’ breath, providing detailed information about its presence [40,101,103]. Similarly, the MQ-138 is designed to detect sulfur dioxide (SO2) and has been successfully used to measure the concentration of this component in patients’ breath, offering valuable data on the quality of inhaled air [40,101,103]. The versatile MQ-135, capable of detecting harmful gases and pollutants, is applied effectively to measure concentrations of various VOCs in patients’ breath, enabling a comprehensive assessment of pollution [40,101,103]. Finally, the AS-MLV (PS-7), specialized in detecting harmful components, has been used to measure concentrations of different VOCs in patients’ breath, consolidating its importance in the accurate detection of substances harmful to health [27,100].

### 4.5. Medical Devices

In this case, 22 medical devices were exclusively identified in medical studies and have been organized into three categories, as detailed in Table 6. Figure 10 presents the devices most frequently identified in the studies, providing a clearer view of their relevance in medical research.

#### 4.5.1. Mass Spectrometry

These devices address various aspects of medical research and contribute significantly to the advancement of mass spectrometry, which is a crucial technique in this discipline. One such device is the Gas Chromatography–Ion Mobility Spectrometry (MSH-5) device, which combines gas chromatography (GC) and ion mobility spectrometry (IMS) [14]. This device was not only used to analyze urine biomarkers, optimizing temperature to maximize information [1], but also proved useful in providing sensitivity and specificity to VOCs in various investigations [68,98,107]. Another notable device is GC Time-of-Flight Mass Spectrometry (MSH-3), which combines GC with time-of-flight mass spectrometry (TOF-MS) [137]. This device played a crucial role in the analysis of VOCs in urine, identification of urinary biomarkers, and distinction between cancer patients and controls [93,98,107,115]. Its purpose is to maximize the separation and chemical content of VOCs [1]. Additionally, the Selected Ion Flow Tube Mass Spectrometry (MMSH-9) is designed for the rapid and real-time analysis of chemical compounds in gaseous or liquid samples [138], and the literature provides information on 116 types of VOCs exhaled by lung cancer patients and healthy subjects [8]. This device also significantly contributed to the identification of carcinogenic and non-carcinogenic groups [10,90,97,98]. Similarly, Gas Chromatography–Mass Spectrometry (MSH-2), used to separate, identify, and quantify chemical compounds in various biomarkers [139], has been widely employed in medical research. This device demonstrated its versatility in analyzing urinary metabolite data, identifying urinary Volatile Organic Compounds, detecting compounds in cancer cells, and analyzing VOCs in breath biomarkers.

#### 4.5.2. Analyte Extraction

This sensor has been pivotal in medical research, and two standout devices have demonstrated their effectiveness in this process. Headspace Solid-Phase Microextraction (EH-2) is a medical device designed for the preconcentration of volatile compounds in liquid or solid biomarkers with application in gas chromatography [64]. This device played a crucial role in optimizing VOCs and in the microextraction of tissue and urine biomarkers [14,76,89,115,116,117]. On the other hand, Solid-Phase Microextraction (EH-5) has excelled in the preconcentration and extraction of volatile and semi-volatile compounds in liquid, solid, or gaseous samples. Its versatility is evident in the detection of components in blood and urine, the extraction of VOCs in the vapor phase, and the microextraction of VOCs in a solid phase.

This device, used independently or in combination with GC-MS, has been instrumental in analyzing VOCs in human breath [5,12,14,15,16,78,79,83]. Both devices have significantly contributed to the efficient extraction of analytes in various medical applications, enhancing the accuracy and quality of the analyses.

As for analytical techniques, the devices MSH-2, MSH-3, MSH-5, and MSH-9 employ analytical techniques such as gas chromatography and mass spectrometry for the analysis of chemical compounds in various biomarkers. These categories represent the fundamental approaches and primary techniques used in medical devices to assess the concentrations of components in the reviewed studies.

### 4.6. Metrics

In the analysis of metrics for assessing odor component detection techniques using electronic nose technologies, certain metrics stand out for their high frequency of occurrence in the reviewed technical and medical literature.

A thorough evaluation of the metrics used to analyze the effectiveness of odor component detection techniques using electronic nose technologies reveals notable patterns in the examined technical and medical literature. Figure 11 highlights that the Area under the Curve emerges as the most frequently cited metric, accounting for 33.47% of the total 236 citations from the reviewed studies. Sensitivity (32.20%) and Specificity (32.20%) closely follow, underscoring their fundamental importance in assessing the models’ ability to detect true positives and true negatives. The Receiver Operating Characteristic (ROC) and Relative Standard Deviations (RSDs) share a representation of 2.54%, contributing to the diversity of metrics used in these studies. Additionally, other metrics such as Negative Predictive Value (NPV) at 2.12%, Limit of Detection (LOD), and *p*-value (both at 1.69%), as well as Confidence Interval (CI) at 1.27%, indicate the variety of aspects addressed in the evaluation of these technologies. Metrics like Area under the ROC (AUROC), Diagnostic Odds Ratio (DOR), Correlation (CRLT), Geometric Mean (G-mean), Recall (RC), Precision (PRE), Real Concentration of Creatinine (R), Root Mean Squared Error of Calibration (RMSEC), Root Mean Square Error of Cross-Validation (RMSECV), Standard Deviation (SD), F1-Score (F1-S), Principal Component (PC), Matthews’s Correlation Coefficient (MCC), Sensor Resistance Ratio (RS), Positive Predictive Value (PPV), and Cross-Validated Accuracy (CVA) have a more limited impact, each representing 0.42% of the citations. Lastly, Retention Time (RT) stands out with 4.66% representation, demonstrating its relevance in the evaluation of these techniques.

## 5. Results


***QSM1:** How do different systematic search methodologies affect the quality and quantity of data retrieved on volatile compounds related to early cancer detection, and what approaches optimize the precision of scientific data collection?* The systematic search methodology employed in this study used the PICOS (see Table 2) approach to define the research questions and structure the search strings within scientific repositories. A total of six repositories were used in the search process, including Scopus, Science Direct, Web of Science, PubMed, IEEE Xplore, and ProQuest. The search strings used for technical studies included “Cancer AND Volatile Organic Compounds AND Techniques AND (Electronic Nose OR eNose OR Sensor),” while for medical studies, the string “(Cancer OR Cancer Cell) AND Volatile Organic Compounds AND (Concentrations OR Biomarkers)” was employed. The application of these search strings improved both the quantity and precision of the retrieved results, identifying 2896 studies for technical studies and 3847 studies for medical studies. To ensure the quality of the selected studies, inclusion and exclusion criteria were applied (see Table 3). These criteria were used to exclude studies that did not meet the required standards, such as those with small sample sizes or inadequate methods. Furthermore, advanced filtering techniques (publications from 2018–2023) and text mining were employed to enhance the precision of the results (see Table 4). The search and selection process yielded 120 relevant studies with an equal distribution between technical and medical studies. The temporal distribution of studies shows an increasing interest in non-invasive screening technologies, especially in 2021. This is significant, as it indicates an increase in the amount of recent research on volatile compounds and early cancer detection.***QSM2:** How has the publication of studies on volatile compound detection for early cancer identification evolved over the last five years, what are the emerging trends in the use of biomedical sensors such as eNose technology, and are there gaps in the literature that limit its development?* The publication of studies on the detection of volatile compounds for the early identification of cancer has shown significant growth over the past five years (see Figure 1), peaking in 2021, with 30% of studies published that year. This increase reflects the growing interest in non-invasive technologies such as eNose technology, which uses biomedical sensors to detect volatile compounds in biological samples such as urine, saliva, breath, and blood. These advances have enhanced the eNose’s ability to identify specific cancer biomarkers quickly, inexpensively and without invasive procedures. Emerging trends include improved eNose sensors capable of detecting compounds at lower concentrations and with greater specificity. The integration of technologies such as artificial intelligence and machine learning has improved the sensitivity and specificity of diagnostic models, facilitating the early detection of cancers such as lung and gastric cancer. In addition, the use of 110 sensors and 22 medical devices in the studies reviewed has advanced segmentation, feature extraction and classification techniques to optimize accuracy. Despite these advances, significant gaps remain, such as the lack of standardization in sensors and the absence of representative databases of cancer-specific biomarkers, which limits the replicability and generalizability of results in clinical settings. These limitations require further research and collaboration to overcome obstacles and advance the clinical implementation of the eNose. Additionally, the selected studies are distributed across journals of varying quartiles, which reflects the quality of the sources in which they were published. A total of 65.83% of the studies came from Q1 journals, which are regarded as having the highest impact and prestige in the field. Meanwhile, 25.83% of the studies were published in Q2 journals, while 5.00% and 0.83% were published in Q3 and Q4 journals, respectively. This distribution suggests that the majority of the selected studies come from highly recognized sources, further reinforcing the quality and relevance of the information used in this systematic review.***QSM3:** What are the primary studies and medical devices used in the detection of VOCs associated with cancer, and how have these devices evolved in terms of technological features and clinical relevance?* The research in the medical field has identified 60 primary studies focused on the use of medical devices to detect VOCs associated with various cancers. These studies emphasize the growing role of technologies such as the eNose in analyzing VOCs present in biological samples like breath, urine, saliva, and blood. The relevant devices identified, detailed in Table 6, have become crucial tools in cancer detection. Over time, these devices have evolved, incorporating advanced sensors capable of detecting VOCs with greater sensitivity and specificity, allowing for the more precise identification of cancer-related biomarkers. This technological evolution has enhanced their ability to detect VOCs in lower concentrations, making these devices increasingly valuable for early-stage cancer detection. These advancements have significantly contributed to the devices’ clinical relevance, improving their applicability in non-invasive cancer diagnostics. However, challenges remain in standardizing these devices across different clinical settings, which is necessary for widespread adoption.***QSR1:** What are the most commonly used machine learning techniques and algorithms in the detection of VOCs associated with cancer, and how do they contribute to the accuracy and efficiency of cancer diagnosis using eNose technology?* SVM, PCA, and ANNs are the most commonly used machine learning techniques in the detection of VOCs associated with cancer using eNose technology. SVM, used in 22 studies, excels in classifying high-dimensional data, improving classification accuracy by effectively separating cancerous from non-cancerous samples. Its robustness to noisy data and ability to handle complex, non-linear relationships enhances early cancer detection with high sensitivity and specificity. PCA, applied in 24 studies, reduces data dimensionality, eliminating noise and improving efficiency, enabling faster processing times critical for clinical decision making. Additionally, PCA improves the signal-to-noise ratio, further increasing diagnostic accuracy. ANNs, utilized in 13 studies, are highly effective in modeling complex patterns, uncovering novel cancer-specific biomarkers by learning from large datasets. Their ability to adapt and improve with new data enhances the detection of subtle interactions between VOCs and cancer types. According to the analysis of the primary literature, there are 34 techniques classified into five categories (see Table 5). Figure 7 illustrates the distribution of 120 citations across these categories, helping to highlight the relative importance of different research techniques in the field. The combined use of these machine learning techniques significantly improves both the accuracy and efficiency of eNose systems, facilitating faster, more reliable cancer diagnoses—essential for early detection and timely interventions in clinical settings.***QSR2:** What are the key VOCs identified as biomarkers for various types of cancer, and how do their concentrations and chemical classifications contribute to early detection and diagnosis?* The concentrations of Volatile Organic Compounds are closely correlated with various types of cancer (see Figure 5), each showing distinct VOC profiles. A comprehensive review identified 661 chemical components across nine types of cancer (see Table A1), categorized into 20 chemical families, which highlight the diagnostic potential of VOCs. Key biomarkers include acetone, ethanol, nonanal, and ammonia, which are strongly associated with lung, colorectal, and bladder cancers. These VOCs provide a non-invasive, efficient approach for early cancer detection, as their concentrations vary significantly between healthy individuals and cancer patients, enabling timely diagnosis. In lung cancer, acetone, ammonia, and ethanol are commonly identified biomarkers, which are present in exhaled breath. For colorectal cancer, alkanes, ketones, and aldehydes are found in urine, blood, and breath, while bladder cancer presents biomarkers such as sulfur compounds, ketones, and hydrocarbons. For prostate cancer, aromatic compounds and alkylbenzenes are key. The categorization of VOCs into chemical families aids in understanding their diagnostic potential, making it easier to pinpoint specific biomarkers for each cancer type. Acetone, particularly for lung cancer, is widely used due to its strong presence in exhaled breath. Similarly, ethyl acetate and toluene have been linked to bladder and lung cancers, respectively. The integration of urine and blood biomarkers enhances the diagnostic accuracy, providing a comprehensive approach to cancer detection. Despite their promise, challenges remain in the clinical implementation of VOC-based diagnostics. Environmental factors, such as temperature and humidity, can cause variability in VOC concentration levels, potentially affecting results. The sensitivity and specificity of VOCs for early-stage cancer detection is still under investigation, as low concentrations of certain biomarkers can hinder detection in the disease’s early stages. The standardization and validation of VOC-based methods are necessary to ensure consistent and reliable results across different clinical settings. For a more complete visualization of how these components are organized and to understand which VOCs are most frequently identified, please take a look at Figure A1, Figure A2, Figure A3, Figure A4, Figure A5, Figure A6 and Figure A7. One of the key advantages of using VOC detection methods for cancer diagnosis includes the non-invasive nature of these techniques, as discussed in reference [88], which facilitates early detection and speeds up the diagnosis process, benefiting patients [16,79]. Additionally, these methods are fast, easy to use, portable, and low cost in practice, providing access to affordable diagnostic devices [86,120]. However, limitations exist, such as the fact that most studies focus on only one type of cancer, as noted in [46], and the high costs associated with the devices used, as mentioned in [101]. Therefore, VOCs show great promise for early cancer detection, offering a non-invasive and efficient approach to diagnosing diseases like lung, colorectal, and bladder cancer, while also highlighting the need for continued research and improvement in detection methods.***QSR3:** What types of sensors have proven most effective for the detection of VOCs in cancer diagnostics, and what are the challenges and limitations in their implementation in clinical settings?* Various types of sensors have been proven effective for detecting VOCs in cancer diagnostics, including metal oxide semiconductor (MOS) sensors, electrochemical sensors, and gas sensors.
-MOS sensors, such as the TGS2610 and TGS2602, are widely used in the detection of VOCs like acetone, ethanol, and ammonia, which are associated with lung, colorectal, and bladder cancers. These sensors are effective at amplifying signals and detecting exhaled VOCs but can be influenced by environmental factors such as temperature and humidity.-Electrochemical sensors, like the MQ-137, MQ-138, and MQ-135, detect gases such as ammonia and sulfur dioxide, which are key biomarkers for lung, colorectal, and bladder cancers. They provide real-time measurements, although their sensitivity to low concentrations of VOCs can limit early-stage detection.-Gas sensors, such as the TGS826 and TGS8669, detect biomarkers like acetone, ethyl acetate, toluene, and ammonia, all of which are crucial for early cancer detection. Acetone is primarily associated with lung cancer, and ethyl acetate with bladder cancer, while toluene has been linked to lung cancer.These sensors provide a non-invasive and reliable approach for cancer detection, especially for early-stage diagnosis. The ability to detect cancer-specific VOCs offers significant advantages in terms of timely diagnosis and intervention.However, there are several challenges in the clinical implementation of these sensors:
-Environmental interference (e.g., temperature and humidity) can affect sensor reliability, causing variations in readings.-Sensitivity limitations in detecting low concentrations of VOCs hinder early detection, particularly in the initial stages of cancer.-High costs associated with specialized sensors and the need for regular calibration are significant barriers for widespread clinical use, especially in resource-limited settings.While these sensors offer great promise, cost-effective and portable diagnostic devices are crucial for large-scale screening, and affordability remains a key challenge. Table A2 and Figure 8 and Figure 9 provide further details on the VOCs detected by these sensors across different cancer types, highlighting the importance of continued advancements in sensor technology.***QSR4:** What are the performance metrics most commonly used to evaluate the effectiveness of eNose technology in detecting cancer-related VOCs, and how do these metrics vary across different studies and cancer types?* The evaluation of eNose technology for detecting cancer-related VOCs commonly uses several key performance metrics: AUC, sensitivity, specificity, accuracy, and precision. These metrics are essential for assessing the ability of eNose systems to discriminate between cancerous and non-cancerous samples as well as for determining their diagnostic performance. Each of these metrics (see Figure 11) is accompanied by corresponding performance data, which is detailed in both the technical and medical literature. For instance, Table A3, Table A4, Table A5 and Table A6 in the technical literature, and Table A7, Table A8, Table A9, Table A10, Table A11 and Table A12 in the medical literature, provide a comprehensive breakdown of results across various categories, including classifiers, segmentation, extraction, mass spectrometry, and regression. These tables present detailed data from technical and medical studies, which help to substantiate the reported performance metrics and offer deeper insights into the practical application of eNose technology.The AUC of the ROC curve is one of the most commonly used metrics to measure how effectively eNose technology can distinguish between cancerous and healthy samples. Higher AUC values indicate greater diagnostic accuracy. In the case of lung cancer, studies often report AUC values above 0.85, reflecting high sensitivity. However, for colorectal cancer and bladder cancer, AUC values tend to be lower, ranging from 0.75 to 0.85, due to the more complex and varied VOC profiles associated with these cancer types. Sensitivity measures the ability of eNose systems to correctly identify cancer patients (true positives). It is a crucial metric for early cancer detection. For lung cancer, sensitivity values range from 80% to 90%, with some studies reaching up to 92%, demonstrating the high effectiveness of the eNose in detecting specific VOCs. In contrast, for colorectal cancer, sensitivity tends to be lower, ranging from 70% to 80%, due to the more subtle nature of the VOC signatures in these cancers. Specificity measures the ability of eNose systems to correctly identify patients without cancer (true negatives), which helps reduce false positives. In lung cancer detection, eNose systems typically report specificity values above 85%, indicating high accuracy in distinguishing between cancerous and non-cancerous VOC profiles. However, specificity tends to be lower in studies on colorectal cancer, with values ranging from 70% to 85%, due to the overlap of VOC profiles from other gastrointestinal diseases. The metrics of accuracy and precision evaluate the overall performance of eNose systems. Accuracy refers to the proportion of correct classifications (both true positives and true negatives), while precision measures the proportion of true positives among all positive predictions. For lung cancer, studies often report accuracy values around 85% to 90% with precision ranging from 75% to 85%. For cancers like gastric or ovarian cancer, accuracy and precision values can fall below 80%, reflecting the more complex and overlapping VOC profiles associated with these cancer types.The performance of eNose technology varies significantly across different cancer types due to the distinct VOC profiles produced by each type of cancer. Lung cancer generally produces the highest values for sensitivity, specificity, and AUC due to well-defined and consistent VOC patterns. In contrast, cancers like ovarian, colorectal, and gastric cancer often present more heterogeneous VOC profiles, resulting in lower sensitivity and specificity values. Early-stage detection is more effective in cancers with clearer VOC markers, such as lung cancer, whereas in other cancers, the VOCs can be more difficult to identify due to less distinct profiles.***QSR5:** Are there any standardized databases or datasets specifically designed for VOCs detection in the context of cancer diagnosis, and how do these databases support the development of more accurate diagnostic tools?* There are several standardized databases specifically designed for the detection of VOCs in the context of cancer diagnosis. These databases are crucial for the development of more accurate diagnostic tools, as they provide standardized datasets for the analysis of VOCs associated with different types of cancer. The Human Metabolome Database contains 220,945 entries of small molecular metabolites present in the human body. It plays a crucial role in cancer diagnosis by providing a comprehensive catalog of metabolites, which facilitates the analysis of biomarkers associated with lung, colorectal, and bladder cancers. The data enable the evaluation of correlations between VOCs and biomarkers, supporting early detection and diagnostic accuracy. The NIST Database is specialized in compound identification, providing reference mass spectra for GC-MS and LC-MS. This database plays a key role in medical research by enabling the identification of VOCs through retention time, fragmentation patterns, and spectral matching, which is essential for identifying cancer biomarkers. The high-quality spectra provided by NIST are crucial for the development of more accurate diagnostic tools. Additionally, the data are utilized in conjunction with modern analytical techniques, such as chemometric analysis and multivariate statistical models, to enhance the precision of VOC identification and improve the reliability of sensor-based diagnostics, facilitating non-invasive early cancer detection through breath analysis. The Neomeditec Database houses detailed information on 327 biomarkers, including their CAS numbers and associations with various cancers. This database is instrumental in evaluating VOCs linked to specific cancers and identifying potential biomarkers for early detection. Its integration with diagnostic technologies such as eNose systems allows for real-time data acquisition and analysis. By combining the biomarker information with sensor data, this database supports the development of precision diagnostic tools that can detect disease-specific VOCs in exhaled breath, thus aiding in the creation of portable, non-invasive devices for cancer diagnosis. The Sigma-Aldrich Alfa Aesar Database offers comprehensive data on VOCs, including their chemical properties, molecular weights, and spectroscopic data. This resource is invaluable for studies aimed at capturing VOCs in both static and dynamic environments, which is essential for understanding VOC behavior in relation to cancer biomarkers. By providing detailed chemical information, it supports the development of accurate sensor systems that can detect trace amounts of VOCs in complex mixtures. Additionally, its data are leveraged in advanced analytical techniques like gas chromatography and sensor fusion, improving the sensitivity and specificity of diagnostic tools, and making them applicable for early-stage cancer detection.


## 6. Discussion

The findings of this study highlight the importance of developing devices for detecting cancer-related odors, particularly through the analysis of VOCs in biological samples. Identifying key components, concentrations, sensors, metrics, cancer types, sample types, medical devices, and techniques provides a comprehensive understanding of the factors involved in early cancer detection and suggests a clear direction for future research. It is essential to advance the development of more sophisticated non-invasive diagnostic devices that can effectively detect VOCs associated with various types of cancer.

The analysis of the reviewed studies reveals that the most commonly used technologies for VOC detection focus on devices with high precision and sensitivity. Among the prominent sensors, GC-MS stands out due to its ability to detect extremely low concentrations of VOCs, ensuring high sensitivity and accuracy. Several studies have shown that GC-MS remains a highly effective technology for identifying specific biomarkers. Another relevant technology is Selected Ion Flow Tube Mass Spectrometry (SIFT-MS), which offers rapid and accurate analysis, with high applicability in clinical settings. On the other hand, metal oxide sensors (MOSs), while more affordable, present certain limitations in clinical environments due to their sensitivity to environmental interference.

Data processing techniques, such as PCA, SVM, and ANN, complement VOC analysis by optimizing signal classification and reducing data dimensionality. These techniques improve diagnostic accuracy and enable the identification of patterns related to different types of cancer. In terms of performance metrics, sensitivity and specificity are key indicators for assessing the effectiveness of the technologies. The use of advanced technologies like GC-MS and machine learning techniques has shown high performance, ensuring accurate non-invasive diagnoses across various types of biological samples, including breath, urine, blood, and saliva. This highlights the potential of these technologies to be used in clinical settings, provided certain limitations are addressed.

Table 7 provides a clear overview of the highlighted technologies, their performance metrics, and their relationship with the analyzed sample types, showcasing key trends and strengths. Sensitivity and specificity metrics are critical for evaluating the effectiveness of these technologies. The integration of advanced devices, paired with analysis techniques such as SVM and ANN, ensures precise non-invasive diagnoses. Notably, these technologies have demonstrated broad applicability across various biological samples, including breath, urine, blood, and saliva.

The results of this study provide a solid foundation for advancing the development of devices and methodologies for cancer detection through VOC analysis. However, there are still technical and operational challenges that must be addressed to maximize the clinical applicability of these technologies. With this in mind, the following recommendations are presented to encourage future development and optimize the implementation of eNose devices in cancer detection:To expand the sample size, it is important to address the limitation of studies with small sample sizes, which restricts the generalizability of the results. Follow-up studies involving larger sample sizes and diverse cohorts are recommended to validate initial findings. Additionally, implementing collaborative studies involving multiple centers could significantly increase the diversity and number of samples, thereby strengthening the validity and robustness of the results obtained.Regarding complementary analytical approaches, it was identified that some methods may have missed important biomarkers. To improve detection, a multidimensional approach combining complementary techniques such as spectroscopy and artificial intelligence analysis is suggested. Furthermore, the application of cross-validation methods can ensure that all relevant biomarkers are identified, improving the accuracy and utility of detection devices.The long-term stability of eNose-based decision models represents a critical challenge. The implementation of recalibration strategies and the use of adaptive algorithms are recommended to maintain accuracy under various conditions and over time. This will ensure that the devices can deliver consistent and reliable results regardless of variations in the environment or conditions of use.To improve sensor sensitivity, it should be considered that some sensors have lower sensitivity to the concentration of VOCs present in the samples, limiting their effectiveness. More selective sensors could be developed, and different types of sensors could be combined to increase detection capacity. Additionally, it is recommended to explore emerging technologies, such as nanostructured materials, which have shown significant potential to improve sensitivity.Cost reduction and operational optimization of the devices represent a challenge for their widespread adoption, given the high costs and the need for trained personnel. Therefore, it is recommended to develop more affordable and portable devices that maintain good performance. Additionally, user-friendly tools and guidelines should be provided to train healthcare professionals, maximizing their adoption and effective use in clinical settings.

## 7. Conclusions

This study contributes significantly to advancing non-invasive diagnostic technologies by highlighting the potential of devices designed to detect cancer-associated VOCs. The identification of key components, sensors, medical devices, and techniques lays the groundwork for advancing non-invasive diagnostic technologies. The unexpected findings, particularly the application of medical devices to analyze VOCs in various sample types, point to significant opportunities for innovation in cancer detection. The integration of these technologies into clinical practice holds the potential to revolutionize early cancer diagnosis, improving survival rates and patient outcomes.

However, several challenges remain, such as the limited availability of public databases and the high costs associated with medical devices. Addressing these issues is essential for making VOC-based detection methods more accessible and widely implemented in clinical settings. Future research should focus on improving the reliability and standardization of these devices while also exploring the use of complementary technologies such as liquid biopsies and medical imaging.

Compounds such as acetone and benzene, detected with high frequency, have the potential to serve as universal biomarkers due to their association with common metabolic pathways in various cancer types, such as altered energy metabolism and oxidative stress. On the other hand, VOCs with moderate recurrence, such as toluene and ethanol, may be linked to specific cancer subtypes or particular stages of the disease. This highlights their role in enabling more precise patient stratification and the personalization of therapeutic strategies. Conversely, low-frequency compounds, such as styrene and butanal, offer diagnostic value for rarer cancers or early-stage detection, as they represent unique metabolic pathways or specific tumor responses. Finally, the high proportion of compounds identified only once underscores the metabolic heterogeneity of cancer and the need for multidimensional approaches to analyze this complexity. While this poses challenges, it also opens avenues for the discovery of novel, clinically relevant chemical signatures.

Finally, the integration of eNose technology into clinical practice represents a promising avenue for early cancer detection, provided that precision is enhanced through advanced machine learning algorithms, calibration protocols are standardized, and the cost-effectiveness of these devices is thoroughly evaluated. Such improvements will contribute to more efficient and accessible cancer diagnosis, ultimately benefiting both healthcare providers and patients. 

## Figures and Tables

**Figure 1 sensors-24-07868-f001:**
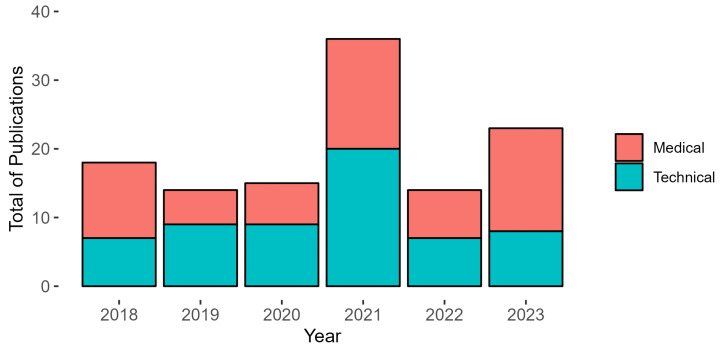
Distribution of annual publications and by type of study.

**Figure 2 sensors-24-07868-f002:**
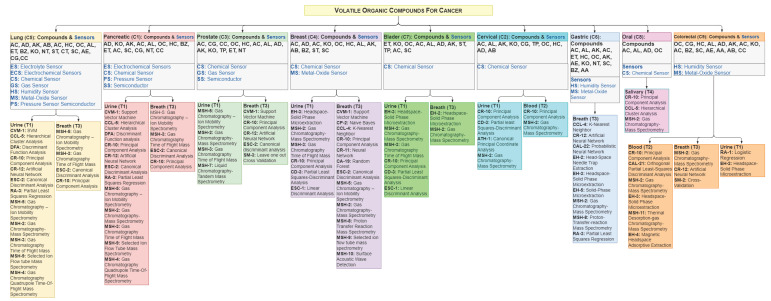
Taxonomy of the Study provides a comprehensive overview of VOCs and their role in cancer detection. It further complements the taxonomic framework by highlighting the classification of VOCs according to cancer types, the biological samples analyzed (breath, urine, saliva, and blood), and the advanced technologies utilized for their detection, such as chemical sensors and semiconductor-based devices. This figure offers a clear perspective on the interplay between chemical biomarkers, diagnostic tools, and analytical technologies, emphasizing the significance of VOCs in early diagnosis and oncological research.

**Figure 3 sensors-24-07868-f003:**
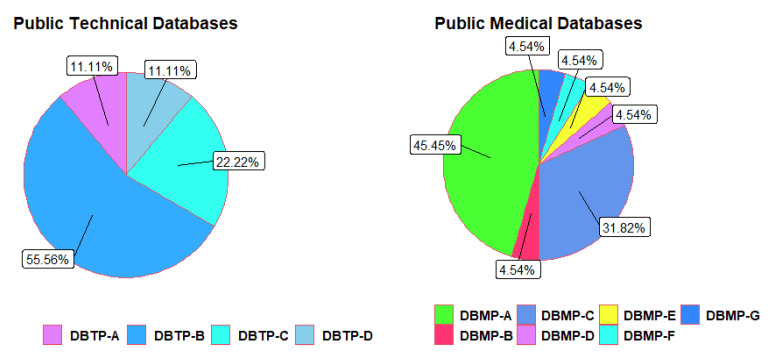
Public database is identified in technical and medical studies.

**Figure 4 sensors-24-07868-f004:**
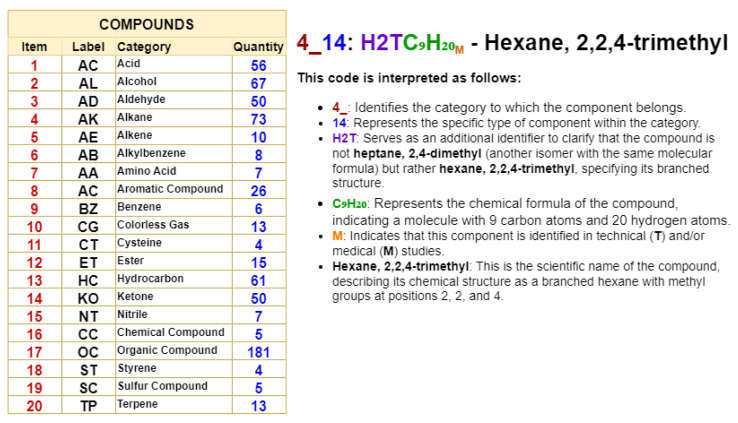
Categories of VOCs and terminology for their identification.

**Figure 5 sensors-24-07868-f005:**
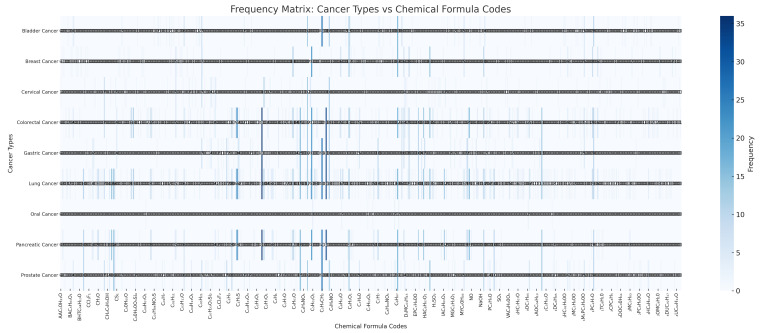
The chart displays a frequency matrix that connects cancer types (vertical axis) with chemical formulas (horizontal axis). Each cell represents the frequency of association between a specific cancer type and a chemical formula, indicated by the intensity of the blue color scale. Rows correspond to specific cancer types, while columns represent coded chemical formulas. This format helps identify patterns and notable relationships, such as formulas highly associated with certain cancer types or cancer types sharing multiple formulas.

**Figure 6 sensors-24-07868-f006:**
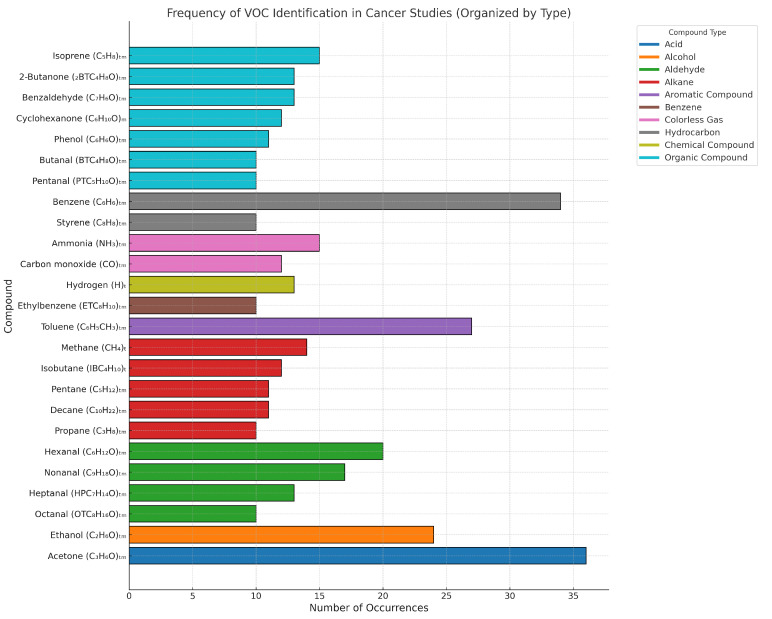
The chart presents the frequency of VOCs identified in technical (T) and medical (M) studies related to cancer, organized by chemical type and occurrence frequency. The horizontal axis displays the total occurrences of each compound, while the vertical axis lists their names and chemical formulas. The compounds are classified into specific chemical categories, visually distinguished by distinct colors assigned to each type, as explained in the legend.

**Figure 7 sensors-24-07868-f007:**
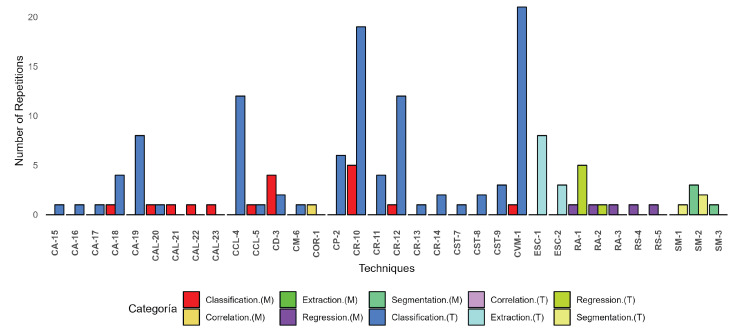
Frequency of identified techniques in the studies and their occurrence in TS and MS.

**Figure 8 sensors-24-07868-f008:**
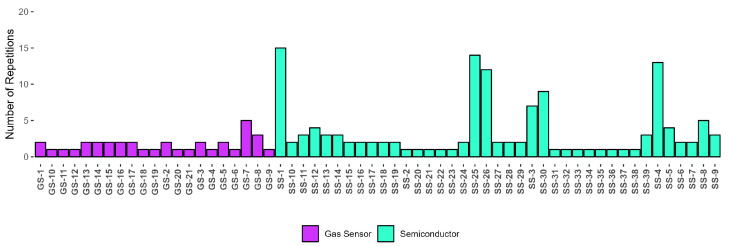
Semiconductor and gas sensors.

**Figure 9 sensors-24-07868-f009:**
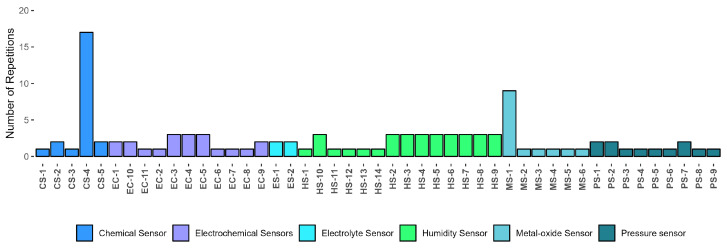
Chemical, electrolyte, metal oxide, electrochemical, pressure, and humidity sensors.

**Figure 10 sensors-24-07868-f010:**
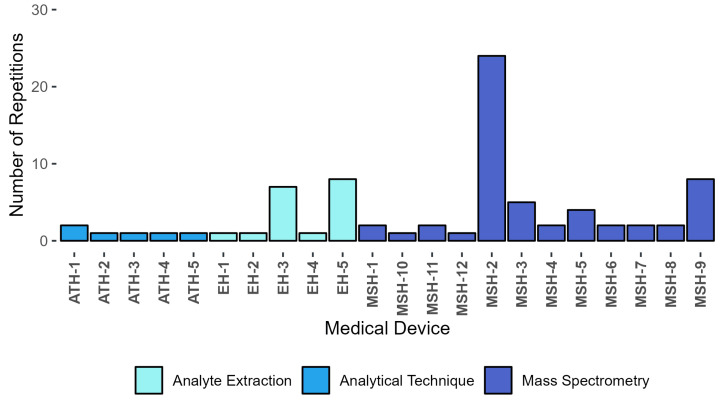
Analytical, chemical method, analyte extraction technique.

**Figure 11 sensors-24-07868-f011:**
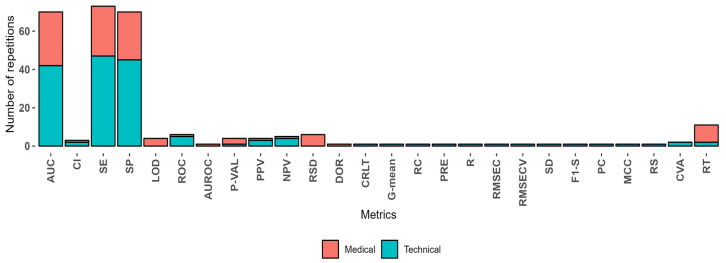
Metrics identified in the technical and medical literature.

**Table 1 sensors-24-07868-t001:** Comparison of eNose performance metrics with traditional diagnostic methods across various cancer types.

Cancer Type	eNose		Traditional			Ref.
	Sensitivity	Specificity	Method ^1^	Sensitivity	Specificity	
Lung	80–90%	80–90%	LDCT	80–90%	80–90%	[4,5,6,7,8,9,10,11,12,13,14,15,16,17,18,19,20,21,22,23,24,25,26,27,28,29,30,31,32,33,34,35,36,37,38,39,40,41,42,43,44,45,46,47,48,49,50,51,52,53,54,55,56,57,58,59,60,61,62,63]
Breast	Up to 90%	80%	MG	75–85%	85–90%	[64,65,66,67,68,69,70,71]
Gastric	85–90%	80–85%	EB	100%	100%	[70,72,73,74,75,76,77,78,79,80,81]
Colon	70–85%	70–85%	CS	>95%	>95%	[82,83,84,85,86,87,88,89,90,91,92,93,94,95]
Pancreas	70–90%	70–90%	CT	70–80%	80–90%	[1,96,97,98,99,100,101,102,103]
Prostate	70–85%	70–85%	PSA	70–80%	60–70%	[104,105,106,107,108,109,110,111,112,113]
Oral	80–90%	70–85%	PE/Bx	40–60%	70–80%	[114]
Bladder	70–85%	70–85%	Cysto/Bx	90–95%	90–95%	[115,116,117,118,119]
Cervical	70–85%	75–85%	Pap Smear	50–80%	85–95%	[120,121,122]

^1^ Method abbreviations: LDCT = low-dose computed tomography, MG = mammography, EB = endoscopy with biopsy, CS = colonoscopy, CT = computed tomography, PSA = prostate-specific antigen, PE/Bx = physical examination/biopsy, Cysto/Bx = cystoscopy with biopsy, Pap Smear = papanicolaou.

**Table 2 sensors-24-07868-t002:** PICOS, inclusion and exclusion criteria for systematic mapping and literature review.

Item ^1^	Considerations	Keywords	Synonyms/Related Terms
**Technical Studies (TS)**			
P	People	Cancer	Cancer Cell OR Cancer Diagnosis
I	Technology/Study tool	Sensors	eNose OR Electronic Nose System
C	Biomarkers	Compounds	Organic Compounds OR Volatile Organic Compounds OR VOCs
O	Performance	Techniques, Metrics	Chemometric Techniques
S	Study Designs	Scientific Literature	Scientific studies
**Medical Studies (MS)**			
P	People	Cancer	Cancer Cell OR Cancer diagnosis
I	Technology/Study tool	Techniques	Chemometric Techniques
C	Measurable Indicators	Concentrations	Organic Compounds OR Volatile Organic Compounds OR VOCs
O	Measurement Units	Concentrations	Biomarkers
S	Study Designs	Medical Literature	Medical studies

^1^ Item = {P: population, I: intervention, C: comparison, O: outcomes, S: study}.

**Table 3 sensors-24-07868-t003:** Inclusion and exclusion criteria for mapping and systematic review of literature.

Inclusion Criteria	Exclusion Criteria
CiSM1: Studies from 2018 to 2023.	CeSM1: Studies without a DOI.
CiSM2: Studies published in English.	CeSM2: Exclude repeated articles.
CiSM3: Synonymous words in the study area.	CeSM3: Exclusion of technical books and conferences.
CiSR1: Inclusion of datasets.	CeSR1: Studies not focused on the detection of odor components in eNose.
CiSR2: Inclusion of sensors and medical device.	CeSR2: Exclude studies lacking metrics, concentrations, components, techniques, and quantitative data.
CiSR3: Inclusion of software.	
CiSR4: Inclusion of cancer concentrations.	
CiSR5: Inclusion of cancer components.	
CiSR6: Inclusion of metrics.	
CiSR7: Quantitative results.	

**Table 4 sensors-24-07868-t004:** Repositories, search strings, and results for technical and medical studies.

Repositories	Search Strings	BDB	F1	F2	F3	F4
**Technical Studies (TS)**						
Scopus	[[Title: Cancer] and [Abstract: Volatile Organic Compounds] AND [Abstract: Electronic Nose] AND [Abstract: Techniques] OR [Abstract: Sensor]]	101	55	50	25	19
Science Direct	[[Title: cancer] and Abstract: Volatile Organic Compounds AND [Abstract: Techniques] AND [Abstract: electronic nose] OR [Abstract: eNose] or [Abstract: Sensor]]	1084	416	72	23	11
Web of Science	[[Title: Cancer] AND [Abstract: Volatile Organic Compounds] and [Abstract: Electronic Nose] OR [Abstract: Sensor]]	390	220	152	35	11
PubMed	[[Title: Cancer] AND [Abstract: Volatile Organic Compounds] OR AND [Abstract: VOCs] AND [Abstract: Techniques] AND [Abstract: Electronic Nose] OR [Abstract: Sensor]]	65	41	25	23	10
ProQuest	[[Title: cancer] AND [Abstract: Volatile Organic Compounds] AND [Abstract: Techniques] AND [Abstract: Electronic Nose] OR [Abstract: Sensor]]	1243	56	34	22	5
IEEE Xplore	[[Title: Cancer] and [Abstract: Volatile Organic Compounds] AND [Abstract: Electronic Nose] OR [Abstract: Sensor]]	13	8	7	5	4
Total		2896	796	340	133	60
**Medical Studies (MS)**						
Web of Science	[[Title: Cancer] AND [Abstract: Volatile Organic Compounds] AND [Abstract: Biomarkers] OR [Abstract: Concentrations]]	369	49	37	35	15
Scopus	[[Title: cancer] OR [Title: Cancer Diagnosis] AND [Abstract: Volatile Organic Compounds] AND [Abstract: Biomarkers]]	332	96	52	27	13
PubMed	[[Title: Cancer] AND [Abstract: Volatile Organic Compound] OR [Abstract: VOCs] AND [Abstract: Biomarkers]]	281	124	57	26	12
Science Direct	[[Title: Cancer] AND [Abstract: Volatile Organic Compounds] OR [Abstract: VOCs] AND [Abstract: Biomarkers]]	1383	416	113	63	12
Springer	[[Title: Cancer] OR [Title: Cancer Diagnosis] AND [Abstract: Volatile Organic Compounds] OR [Abstract: Organic Compounds] AND [Abstract: Concentrations] OR [Abstract: Biomarkers]]	330	127	36	21	4
ProQuest	[[Title: Cancer] OR [Title: Cancer Cell] AND [Abstract: Volatile Organic Compounds] AND [Abstract: Concentrations] OR [Abstract: Biomarkers]]	1152	368	143	58	4
Total		3847	1180	438	230	60

**Table 5 sensors-24-07868-t005:** Segmentation, extraction, regression, correlation, and classification techniques identified in TS and MS.

Label: Technical Description	Cancer ^1^	Ref
**Segmentation**		
SM-1: K-fold Cross-Validation	C5	[43]
SM-2: Cross-Validation	C5, C9	[5,7,55,59,92]
SM-3: Leave One Out Cross-Validation	C3	[109]
**Correlation**		
COR-1: Canonical Correlation Analysis	C3	[21]
**Extraction**		
ESC-1: Linear Discriminant Analysis	C5, C7	[24,30,34,39,40,50,55,119]
ESC-2: Canonical Discriminant Analysis	C1, C3, C4	[71,99,111]
**Regression**		
RA-1: Logistic Regression	C5, C9	[31,34,50,52,60,84]
RA-2: Logistic Regression Analysis	C5	[14,48]
RA-3: Partial Least Squares Regression	C1, C6	[72,101]
RS-4: Lasso Regression	C5	[7]
RS-5: Multivariable Logistic Regression	C5	[6]
**Classification**		
CVM-1: Support Vector Machine	C5, C1, C3, C4	[23,28,29,30,31,33,34,37,39,40,41,50,51,52,54,59,63,70,71,101,103,109]
CP-2: Naive Bayes	C5, C4	[23,28,30,34,40,70]
CD-3: Partial Least Squares–Discriminant Analysis	C5, C2, C7	[5,14,64,119,120,122]
CCL-4: K-Nearest Neighbor	C5, C6, C4	[23,29,30,34,39,40,45,50,59,63,70,80]
CCL-5: Hierarchical Cluster Analysis	C8, C1	[101,114]
CM-6: Gradient Boosting	C5	[40]
CST-7: LogitBoost	C5	[23]
CST-8: Discriminant Function Analysis	C5, C1	[25,103]
CST-9: AdaBoost	C5	[37,42,43]
CR-10: Principal Component Analysis	C2, C9, C8, C3, C5, C7, C4, C1	[25,32,33,39,46,50,53,54,58,59,61,69,71,83,99,101,103,109,111,114,119,120,121,122]
CR-11: Neural Network	C5, C4	[33,52,63,70]
CR-12: Artificial Neural Network	C5, C3, C6, C1, C9	[22,27,31,32,36,45,50,52,63,81,95,100,110]
CR-13: Fx-ConvNet	C5	[40]
CR-14: Kernel Principal Component Analysis	C5	[26,58]
CA-15: Decision Tree Algorithm	C5	[40]
CA-16: CatBoost	C5	[40]
CA-17: Light Gradient Boosting Machine	C5	[40]
CA-18: eXtreme Gradient Boosting	C5	[8,31,40,42,51]
CA-19: Random Forest	C5, C4	[31,40,42,50,52,56,60,70]
CAL-20: Discriminant Analysis	C5	[22,47]
CAL-21: Orthogonal Partial Least-Squares Discriminant Dnalysis	C9	[83]
CAL-22: Probabilistic Neural Network	C6	[75]
CAL-23: Gradient-Boosted Decision Trees	C5	[22]

^1^ Cancer = {C1: pancreatic cancer, C2: cervical cancer, C3: prostate cancer, C4: breast cancer, C5: lung cancer, C6: gastric cancer, C7: bladder cancer, C8: oral cancer, C9: colorectal cancer}.

**Table 6 sensors-24-07868-t006:** Medical devices.

Label: Medical Devices	Category VOCs ^1^	Cancer ^2^	Test ^3^	Ref
**Mass Spectrometry**				
MSH-1: Gated Recurrent Unit-Based Autoencoder	AC, CG, CC, OC, HC, AC, AL, AD, AK, KO, TP, ET, NT	C5	T3	[58,59]
MSH-2: Gas Chromatography–Mass Spectrometry	AC, AL, AK, KO, CG, TP, OC, HC, AD, AB	C1, C2, C3, C4, C5, C6, C7, C8, C9	T1, T2, T3, T4	[10,15,19,20,35,38,59,62,65,67,68,73,74,75,83,88,90,92,96,104,105,106,112,113,114,116,117,120,121]
MSH-3: Gas Chromatography–Time of Flight Mass	AD, KO, AK, AC, AL, OC, HC, BZ, ET, AC, SC, CG, NT, CC	C1, C7, C3, C9	T1, T3	[1,93,98,107,115]
MSH-4: Gas Chromatography Quadrupole Time-Of-Flight Mass Spectrometry	AD, KO, AK, AC, AL, OC, HC, BZ, ET, AC, SC, CG, NT, CC	C5, C1	T3	[59,101]
MSH-5: Gas Chromatography–Ion Mobility Spectrometry	AD, KO, AK, AC, AL, OC, HC, BZ, ET, AC, SC, CG, NT, CC	C1, C3, C4	T1, T3	[1,11,68,107]
MSH-6: Gas Chromatography–High-Resolution Mass Spectrometry	AC, AD, AC, KO, OC, HC, AL, AK, AB, BZ, ST, SC	C4	T1	[66,121]
MSH-7: Liquid Chromatography–Tandem Mass Spectrometry	AC, CG, CC, OC, HC, AC, AL, AD, AK, KO, TP, ET, NT	C5, C3	T1, T3	[11,108]
MSH-8: Proton Transfer Reaction Mass Spectrometry	AC, AL, AK, AC, ET, HC, OC, AK, AE, KO, NT, SC, BZ, AA	C4, C5	T3	[44,68]
MSH-9: Selected Ion Flow Tube–Mass Spectrometry	AD, KO, AK, AC, AL, OC, HC, BZ, ET, AC, SC, CG, NT, CC	C1, C5, C4, C9	T1, T3	[8,10,16,68,90,91,97,98]
MSH-10: Surface Acoustic Wave Detection	AC, AD, AC, KO, OC, HC, AL, AK, AB, BZ, ST, SC	C1, C4	T3	[67,102]
MSH-11: Thermal Desorption–Gas Chromatography–Mass Spectrometry	OC, CG, HC, AL, AD, AK, AC, KO, AC, BZ, SC, AE, AA, AB, CC	C5, C9	T2, T3	[9,82]
MSH-12: Proton-Transfer-Reaction Mass Spectrometry	AC, AL, AK, KO, CG, TP, OC, HC, AD, AB	C4	T3	[77]
**Analyte Extraction**				
EH-1: Direct Immersion	AC, AD, AC, KO, OC, HC, AL, AK, AB, BZ, ST, SC	C5	T3	[14]
EH-2: Headspace Needle Trap Extraction	AC, AD, AC, KO, OC, HC, AL, AK, AB, BZ, ST, SC	C8	T3	[74]
EH-3: Headspace Solid-Phase Microextraction	AC, AL, AK, AC, ET, HC, OC, AK, AE, KO, NT, SC, BZ, AA	C4, C7, C5, C7, C9	T1, T3	[14,64,76,89,115,116,117]
EH-4: Magnetic Headspace Adsorptive Extraction	OC, CG, HC, AL, AD, AK, AC, KO, AC, BZ, SC, AE, AA, AB, CC	C9	T2	[82]
EH-5: Solid-phase Microextraction	AC, AL, AK, AC, ET, HC, OC, AK, AE, KO, NT, SC, BZ, AA	C5, C7, C9	T2, T3	[5,12,14,15,16,78,79,83]
**Analytical Technique**				
ATH-1: Canonical Principal Coordinate Analysis	AC, AL, AK, KO, CG, TP, OC, HC, AD, AB	C2, C9	T1, T2	[85,120]
ATH-2: Fourier Transform Infrared	OC, CG, HC, AL, AD, AK, AC, KO, AC, BZ, SC, AE, AA, AB, CC	C5	T3	[13]
ATH-3: Gas Chromatography	AC, AD, AC, KO, OC, HC, AL, AK, AB, BZ, ST, SC	C4	T3	[67,68]
ATH-4: Gas Chromatography with Flame Ionization Detector	AC, AL, AK, KO, CG, TP, OC, HC, AD, AB	C5	T3	[20]
ATH-5: Solid Phase Microextraction - Gas Chromatography	AC, AL, AK, AC, ET, HC, OC, AK, AE, KO, NT, SC, BZ, AA	C5	T3	[13]

^1^ Category VOCs = {AC: acetate, CG: glucose chloride, CC: carbon chloride, OC: octane, HC: hexane, AL: alkanes, AD: acid, AK: acetone, KO: kerosene, TP: trifluoride, ET: ethanol, NT: nitrogen, AB: boric acid, BZ: benzene, ST: toluene, SC: squalene acetate, AA: acetic acid, AE: ethyl acetate}; ^2^ Cancer = {C1: pancreatic cancer, C2: cervical cancer, C3: prostate cancer, C4: breast cancer, C5: lung cancer, C6: gastric cancer, C7: bladder cancer, C8: oral cancer, C9: colorectal cancer}; ^3^ Test = {T1: urine, T2: blood, T3: breath, T4: saliva}.

**Table 7 sensors-24-07868-t007:** Summary of technologies, performance metrics, and sample types.

Sensors	Sample Type	Concentrations (VOCs)	Detection Techniques	Performance (Metrics)
GC-MS	Breath	0.1 ppm	GC-MS	High Sensitivity, 95%
SIFT-MS	Urine	0.05 ppm	PCA	High Specificity, 90%
MOS	Blood	0.3 ppm	ANN	Medium Sensitivity, 80%
Chemiresistor	Saliva	0.2 ppm	SVM	High Precision, 92%
GC-IMS	Breath	0.15 ppm	PCA	High Sensitivity, 93%

## Data Availability

No new data were created or analyzed in this study. Data sharing is not applicable to this article as it is based on a systematic review of existing literature.

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
