# Peer review of "Exploring Components, Sensors, and Techniques for Cancer Detection via eNose Technology: A Systematic Review"

_sensors, 2024, doi:10.3390/s24237868_

Round 1

Reviewer 1 Report

Comments and Suggestions for Authors

although the authors present a detailed analysis of biomarkers and sensing technology, the biomarkers for different cancers aren't commonly recognized, and the reliability of different detections should be discussed. this discussion should be careful and serious. hopefully, the authors can give an in-depth discussion.

Author Response

Comments 1: Although the authors present a detailed analysis of biomarkers and sensing technology, the biomarkers for different cancers aren't commonly recognized, and the reliability of different detections should be discussed. this discussion should be careful and serious. hopefully, the authors can give an in-depth discussion.

Response 1:

The reliability of detections made by eNose devices for cancer biomarkers has been extensively analyzed in terms of performance metrics such as sensitivity and specificity.

Sensitivity measures the ability of the device to correctly identify individuals with the disease (true positives), whereas specificity indicates its ability to correctly identify those without the disease (true negatives). Table 1 compares eNose detection metrics with traditional methods across various cancer types, demonstrating comparable and promising results.

Table 1. Comparison of eNose Performance Metrics with Traditional Diagnostic Methods Across

Various Cancer Types

CancerType

eNose

Traditional

Ref.

Sensitivity

Specificity

Method1

Sensitivity

Specificity

Lung

80-90%

80-90%

LDCT

80-90%

80-90%

[4–63]

Breast

Up to 90%

80%

MG

75-85%

85-90%

[64–71]

Gastric

85-90%

80-85%

EB

100%

100%

[70,72–81]

Colon

70-85%

70-85%

CS

>95%

>95%

[82–95]

Pancreas

70-90%

70-90%

CT

70-80%

80-90%

[1,96–103]

Prostate

70-85%

70-85%

PSA

70-80%

60-70%

[104–113]

Oral

80-90%

70-85%

PE/Bx

40-60%

70-80%

[114]

Bladder

70-85%

70-85%

Cysto/Bx

90-95%

90-95%

[115–119]

Cervical

70-85%

75-85%

Pap Smear

50-80%

85-95%

[120–122]

1 Method={LDCT: Low-Dose Computed Tomography, MG: Mammography, EB: Endoscopy with Biopsy, CS: Colonoscopy, CT: Computed Tomography, PSA: Prostate-Specific Antigen, PE/Bx: Physical Examination / Biopsy, Cysto/Bx: Cystoscopy with Biopsy, Pap Smear: Papanicolaou}

This comparative analysis indicates that both traditional diagnostic methods and eNose detection provide acceptable levels of reliability, as evidenced by the reported sensitivity and specificity percentages for various cancer types. eNose detection emerges as a promising and favorable alternative, particularly due to its non-invasive and potentially accessible nature, making it an attractive tool for early cancer detection. However, despite its advances and encouraging results, it is important to emphasize that eNose detection should be supplemented with traditional techniques, such as biopsies or imaging diagnostics, to confirm diagnoses and achieve definitive results. While eNose detection does not replace traditional methods, its development and application represent a significant step toward creating more accessible and less invasive diagnostic tools. The ability to efficiently and non-invasively identify cancer biomarkers opens new opportunities for screening and patient monitoring, thereby enhancing early detection and clinical outcomes.

Notes:

·        In subsection 4.2. Analysis of Volatile Organic Compounds, a new subsection, 4.2.1. VOC-Based Biomarkers for Early Cancer Detection Using eNose Technology (Pages 11–15), has been added, focusing on the sensitivity and specificity of eNose technology in detecting unique VOC signatures for early cancer detection. Scientifically grounded updates enhance the discussion of its potential diagnostic applications. An in-depth analysis of 661 identified components has also been conducted, reorganizing the most frequently detected VOCs. These are now detailed in subsection 4.2.2. Metabolic Patterns and Potential Biomarkers in the Identification of VOCs Associated with Cancer (Pages 16–21). This section highlights metabolic pathways and recurring VOC patterns, categorizing VOCs by chemical groups (e.g., hydrocarbons, alcohols, aldehydes) and their correlations with cancer types, improving clarity and diagnostic relevance. These updates strengthen the section's scientific rigor and practical value, offering valuable insights for advancing VOC-based cancer detection techniques and clinical applications.

·        To enhance document comprehension, a dedicated section has been created specifically to organize extensive tables. This approach ensures a cleaner and more concise presentation of the main content. Appendix A: Organize Components by Cancer Type (Pages 37-40), Appendix B: Key Sensors and Their Associated Components in Odor Detection (Pages 41-43), and Appendix C: Results of Technical Studies: Classifiers (Pages 44-52). Similarly, although the possibility of reducing or removing these tables was considered, it was decided to retain them due to the detailed and relevant information they provide regarding the findings identified in the study.

·        Although the possibility of reducing or removing these tables was considered, they were retained because they contain detailed and relevant information about the findings presented in the study.

·        The abstract and introduction were revised to emphasize their scientific relevance.

·        Subsection 4.2.1, VOC-Based Biomarkers for Early Cancer Detection Using eNose Technology (pages 13 and 14), was added. This subsection provides a summary of the effectiveness of this technology in identifying various types of cancer through the analysis of VOCs in different biological samples, including blood, breath, saliva, and urine.

·        Here is a summary of how each question contributed to the findings and scientific contribution of the paper:

1.      QSM1: How do different systematic search methodologies affect the quality and quantity of data retrieved on volatile compounds related to early cancer detection, and what approaches optimize the precision of scientific data collection?
In the Results section, we highlighted how the use of systematic search methodologies, such as the PICOS approach, improved the precision and quantity of data retrieved, particularly in identifying relevant VOCs related to cancer. This is an important contribution to the field, showing how methodological rigor impacts research outcomes (Page 30).

2.      QSM2: How has the publication of studies on volatile compound detection for early cancer identification evolved over the last five years, what are the emerging trends in the use of biomedical sensors such as eNose technology, and are there gaps in the literature that limit its development?
We examined the evolution of research in VOC detection over the last five years, revealing trends such as the growing adoption of non-invasive diagnostic tools like eNose. However, the results also pointed out critical gaps, particularly the lack of standardized sensor technologies and limited clinical validation, which restricts the broader adoption of eNose devices (Page 30).

3.      QSM3: What are the primary studies and medical devices used in the detection of VOCs associated with cancer, and how have these devices evolved in terms of technological features and clinical relevance?
The Results section discusses the evolution of key medical devices, including gas chromatography-mass spectrometry and ion flow tube mass spectrometry, and their integration with eNose technology. The review shows the advancements in sensor sensitivity and the increasing use of machine learning algorithms to enhance diagnostic accuracy (Page 31).

4.      QSR1: What are the most commonly used machine learning techniques and algorithms in the detection of VOCs associated with cancer, and how do they contribute to the accuracy and efficiency of cancer diagnosis using eNose technology?
We identified that Support Vector Machines, Principal Component Analysis, and Artificial Neural Networks are the most commonly used machine learning techniques in this field. The Results section presents a critical analysis of how these algorithms improve the detection accuracy of VOCs in cancer diagnosis, particularly in distinguishing between different types of cancer (Page 31).

5.      QSR2: What are the key VOCs identified as biomarkers for various types of cancer, and how do their concentrations and chemical classifications contribute to early detection and diagnosis?
We found that several VOCs, such as acetone, ammonia, and toluene, are consistently identified as biomarkers for different cancers. The Results section discusses the role of these VOCs in early cancer detection, focusing on their concentrations and how they contribute to non-invasive diagnostic methods (Page 31).

6.      QSR3: What types of sensors have proven most effective for the detection of VOCs in cancer diagnostics, and what are the challenges and limitations in their implementation in clinical settings?
The results show that sensors based on metal oxide semiconductors (MOS) and polymer composites are the most effective for VOC detection. However, challenges such as sensor stability, reproducibility, and the integration of these sensors into clinical settings are critically discussed in the Results section (Page 32).

7.      QSR4: What are the performance metrics most commonly used to evaluate the effectiveness of eNose technology in detecting cancer-related VOCs, and how do these metrics vary across different studies and cancer types?

Los resultados destacan la eficacia de eNose en identificar perfiles VOC específicos, relacionados con el cáncer utiliza métricas como AUC, sensibilidad, especificidad, precisión y exactitude (Page 33).

8.      QSR5: Are there any standardized databases or datasets specifically designed for VOCs detection in the context of cancer diagnosis, and how do these databases support the development of more accurate diagnostic tools?
We identified several public and private databases, such as the Human Metabolome Database (HMDB) and NIST Database, which are crucial for supporting research in VOC detection. The Results section analyzes these databases' importance in improving cancer diagnostics' accuracy and consistency through standardized datasets (Page 33).

Reviewer 2 Report

Comments and Suggestions for Authors

I think the review has obvious in content and design. The present form is just an compile of related papers. No meaningful analysis and comparison are found in the paper.

Firstly, the abstract only provides information about number of topic, publications in the webof science. No content of scientific importance are found.

Secondly, the paper is full of tables that is only an assembly of related information. It is also impossible for reader to read through the trivial details.

Thirdly, the questions given by the authors, QSM1 to PSR5, as seen in page 5, are mostly very superficial. It is like an paper investigation from statistic view, but not a comprehensive review. Because no any analytic comments or cirtical review.

All in all, the paper is very badly structured and can provide few scientific help for the related researchers. Basically the analytical function of webofscience can do similar work. Thus, it should be rejected.

Author Response

Response 1: Thank you for your valuable comments. We would like to clarify that this article is a systematic review and not an original research article. The main purpose of a review article is not to present new scientific findings, but to synthesize and analyze existing research in a specific field.

The abstract, as you pointed out, mentions the number of studies and the publication sources (such as Web of Science), which is a typical feature of a systematic review, it is also important to mention that the entire content of the abstract was reworded

This is in line with the purpose of summarizing the current state of research on eNose technology for cancer detection, rather than presenting new experimental data. As stated in the abstract (page 1): "This paper offers a systematic review of advancements in electronic nose (eNose) technologies for early cancer detection, with a particular focus on the detection and analysis of volatile organic compounds (VOCs) present in biomarkers such as breath, urine, saliva, and blood."

This overview provides the necessary context for the subsequent detailed analysis of the reviewed studies, which is the essence of a systematic review. Moreover, the abstract also emphasizes the inclusion of both technical and clinical perspectives, which are central to the goal of synthesizing existing research (Page 1): "Our review synthesizes both technical and clinical perspectives, evaluating sensor-based devices such as gas chromatography-mass spectrometry and selected ion flow tube mass spectrometry."

Additionally, the methodology used in this paper involves a systematic mapping and literature review process, where 120 studies published between 2018 and 2023 were analyzed using the PICOS methodology (Page 5, Section 3.1): "A total of 120 studies published between 2018 and 2023 were examined through systematic mapping and literature review methodologies, employing the PICOS methodology to guide the analysis."

This approach highlights the paper's role in analyzing and summarizing existing research, as is customary for review articles.

Thus, while the abstract does not present original scientific results, it effectively summarizes the comprehensive analysis of existing literature, which is the main purpose of a systematic review article. Should further clarification be needed, we are happy to enhance the discussion on the contributions of the reviewed studies throughout the manuscript.

Comments 2: Secondly, the paper is full of tables that is only an assembly of related information. It is also impossible for reader to read through the trivial details.

Response 2: Thank you for your insightful feedback. We understand your concern regarding the extensive use of tables and the potential challenge they may pose to readers in terms of accessibility and readability.

As this paper is a systematic review, the primary objective is to present a comprehensive synthesis of the existing literature. The tables serve to organize and clearly present a large amount of data from the studies we reviewed, such as the various components, biomarkers, and techniques used in the detection of volatile organic compounds related to cancer. These tables are intended to provide an overview of the findings from multiple studies, offering readers a clear and structured summary of the key information.

However, we recognize that the tables may appear dense or overwhelming to some readers. In light of this, we are open to adjusting their presentation to improve readability. We have moved the extensive and detailed tables to the appendices, allowing the main body of the text to focus on the more critical aspects of the review. Additionally, we have conducted a thorough review of the chemical formulas, as the subscript in these was found to be incorrect. The extensive tables are now organized in Appendix A: Organize Components by Cancer Type (Pages 35-40), Appendix B: Key Sensors and Their Associated Components in Odor Detection (Pages 41-43), and Appendix C: Results of Technical Studies: Classifiers (Pages 44-53). This allows readers to refer to the appendices for in-depth details when necessary.

We believe that these adjustments could help address your concern while maintaining the comprehensive nature of the review. Please let us know if these changes meet your expectations or if there are specific sections where you feel the tables could be further simplified.

Comments 3: Thirdly, the questions given by the authors, QSM1 to PSR5, as seen in page 5, are mostly very superficial. It is like an paper investigation from statistic view, but not a comprehensive review. Because no any analytic comments or cirtical review.

Response 3: Thank you for your valuable feedback. We would like to address your concerns regarding the questions posed by the authors (QSM1 to PSR5) and the depth of the analysis in this paper.

First, it is important to clarify that the questions were specifically designed to guide the systematic mapping and literature review processes. These questions were reformulated and are discussed in the Results section (Pages 30-34). The purpose of these questions was to structure the review and identify trends, gaps, and key findings in the existing research on the detection of volatile organic compounds (VOCs) for cancer diagnosis using eNose technology.

Here is a summary of how each question contributed to the findings and scientific contribution of the paper:

1.      QSM1: How do different systematic search methodologies affect the quality and quantity of data retrieved on volatile compounds related to early cancer detection, and what approaches optimize the precision of scientific data collection?
In the Results section, we highlighted how the use of systematic search methodologies, such as the PICOS approach, improved the precision and quantity of data retrieved, particularly in identifying relevant VOCs related to cancer. This is an important contribution to the field, showing how methodological rigor impacts research outcomes (Page 30).

2.      QSM2: How has the publication of studies on volatile compound detection for early cancer identification evolved over the last five years, what are the emerging trends in the use of biomedical sensors such as eNose technology, and are there gaps in the literature that limit its development?
We examined the evolution of research in VOC detection over the last five years, revealing trends such as the growing adoption of non-invasive diagnostic tools like eNose. However, the results also pointed out critical gaps, particularly the lack of standardized sensor technologies and limited clinical validation, which restricts the broader adoption of eNose devices (Page 30).

3.      QSM3: What are the primary studies and medical devices used in the detection of VOCs associated with cancer, and how have these devices evolved in terms of technological features and clinical relevance?
The Results section discusses the evolution of key medical devices, including gas chromatography-mass spectrometry and ion flow tube mass spectrometry, and their integration with eNose technology. The review shows the advancements in sensor sensitivity and the increasing use of machine learning algorithms to enhance diagnostic accuracy (Page 31).

4.      QSR1: What are the most commonly used machine learning techniques and algorithms in the detection of VOCs associated with cancer, and how do they contribute to the accuracy and efficiency of cancer diagnosis using eNose technology?
We identified that Support Vector Machines, Principal Component Analysis, and Artificial Neural Networks are the most commonly used machine learning techniques in this field. The Results section presents a critical analysis of how these algorithms improve the detection accuracy of VOCs in cancer diagnosis, particularly in distinguishing between different types of cancer (Page 31).

5.      QSR2: What are the key VOCs identified as biomarkers for various types of cancer, and how do their concentrations and chemical classifications contribute to early detection and diagnosis?
We found that several VOCs, such as acetone, ammonia, and toluene, are consistently identified as biomarkers for different cancers. The Results section discusses the role of these VOCs in early cancer detection, focusing on their concentrations and how they contribute to non-invasive diagnostic methods (Page 31).

6.      QSR3: What types of sensors have proven most effective for the detection of VOCs in cancer diagnostics, and what are the challenges and limitations in their implementation in clinical settings?
The results show that sensors based on metal oxide semiconductors (MOS) and polymer composites are the most effective for VOC detection. However, challenges such as sensor stability, reproducibility, and the integration of these sensors into clinical settings are critically discussed in the Results section (Page 32).

7.      QSR4: What are the performance metrics most commonly used to evaluate the effectiveness of eNose technology in detecting cancer-related VOCs, and how do these metrics vary across different studies and cancer types?

Los resultados destacan la eficacia de eNose en identificar perfiles VOC específicos, relacionados con el cáncer utiliza métricas como AUC, sensibilidad, especificidad, precisión y exactitude (Page 33).

8.      QSR5: Are there any standardized databases or datasets specifically designed for VOCs detection in the context of cancer diagnosis, and how do these databases support the development of more accurate diagnostic tools?
We identified several public and private databases, such as the Human Metabolome Database (HMDB) and NIST Database, which are crucial for supporting research in VOC detection. The Results section analyzes these databases' importance in improving cancer diagnostics' accuracy and consistency through standardized datasets (Page 33).

While these questions might appear to be introductory, they play an essential role in synthesizing and framing the results in a broader context. The Results section (starting on Page 28) provides a more analytical approach, offering critical insights into the scientific contributions from the reviewed studies, such as the development of advanced sensors and machine learning algorithms for better diagnosis, as well as the challenges faced by the field, including sensor standardization and data integration.

Therefore, while the questions themselves serve as a framework, the Results section provides an in-depth and critical evaluation of the findings, highlighting the scientific contributions and areas that require further exploration.

We hope that this clarification helps to address your concerns, and we would be happy to further refine the analysis or expand on any specific points if needed.

Comments 4: All in all, the paper is very badly structured and can provide few scientific help for the related researchers. Basically the analytical function of webofscience can do similar work. Thus, it should be rejected.

Response 4: Thank you for your feedback. We understand your concerns regarding the structure of the paper and its perceived limited contribution to scientific research. However, we would like to address these points in detail to clarify the paper’s intended structure and its value to the field.

1.      Paper Structure: While we acknowledge that the paper may initially appear to have a complex structure due to the extensive data and tables, the intention was to provide a systematic review that offers a comprehensive and organized analysis of the current state of research in the field of electronic nose (eNose) technology for cancer detection. The structure of the paper follows a clear logical flow, starting with an introduction to the technology, followed by detailed sections on background, research questions, methodology, results, and conclusions. This is typical of a review article aimed at synthesizing and critically analyzing the literature.

  • Introduction and Background: These sections lay the foundation for the research by introducing eNose technology and its potential applications in cancer detection (Pages 1-6).
  • Research Questions and Methodology: The research questions (QSM1 to PSR5) and methodology used to conduct the systematic mapping and literature review are presented clearly, guiding the reader through the scientific inquiry (Page 5 onwards).
  • Results and Critical Analysis: The results are not only descriptive but include critical insights into the gaps, challenges, and emerging trends in the field. For example, we highlight the limitations of current sensor technologies, such as the need for standardization, and discuss how the integration of machine learning could enhance diagnostic capabilities (Pages 28-33).

2.      Scientific Contribution: While tools like Web of Science can provide data on the number of publications and trends in the field, this paper goes beyond mere data aggregation. It offers a critical synthesis of the existing literature, providing in-depth analysis and discussion on:

  • The technological advancements and challenges in eNose systems.
  • The clinical applications of eNose for early cancer detection.
  • The limitations of current methods and the gaps that future research must address to improve diagnostic accuracy and clinical relevance.

For example, in the Results section, we provide a detailed evaluation of 120 studies published between 2018 and 2023, with an emphasis on VOCs detected in cancer diagnosis, the types of sensors used, and the machine learning algorithms that have been applied to enhance detection accuracy (Pages 28-32). This critical review is more than just data aggregation; it provides actionable insights for researchers in the field to address ongoing challenges and explore new directions for innovation.

3.      Scientific Help to Researchers: The goal of this review article is to provide researchers with a knowledge base that can guide future studies and innovations in eNose technology for cancer detection. By summarizing the state of the art and identifying research gaps, we hope to offer a resource that is both valuable and informative to those working in the field. For instance, the analysis of the most commonly used machine learning techniques in VOC detection and the challenges of integrating sensors into clinical practice provides researchers with clear guidance on where future efforts are needed (Pages 30-33).

While we appreciate your perspective, we believe that the paper offers significant scientific value by synthesizing the vast body of literature on eNose technology and providing a critical analysis of the challenges and opportunities in the field. This is more than what can be achieved by simply using database tools like Web of Science, as our review adds expert commentary and a structured analysis that would be beneficial for researchers aiming to advance the field.

We are open to further suggestions on how to improve the paper’s structure or its contribution to the scientific community, and we hope this response helps clarify the intent and value of the work presented.

Reviewer 3 Report

Comments and Suggestions for Authors

The scientific review “Exploring Components, Sensors, and Techniques for Cancer Detection via eNose Technology: A Systematic Review” is devoted to a systematic review of the literature on analyzing patient exhalation for cancer diagnosis. The study contains a tremendous amount of diverse data. The authors have done a lot of work. After some modifications, it may be published in "Sensors".

1) The work is very extensive and contains a large number of sections, graphs and pages. I would recommend preparing a table of contents at the beginning of the paper, with the ability to quickly jump to the necessary section.

2) When abbreviations are mentioned for the first time, it is necessary to give their full name. For example, PICOS, etc.

3) Many tables and graphs should be accompanied by a legend or explanations of what the abbreviations used mean. Now their decoding is given in the text, and it is not convenient when analyzing tables and graphs. I recommend that wherever there are abbreviations to make appropriate deciphering.

4) The authors have provided unique and very useful data on various markers in section 4.2. Brief summaries are provided for each of them. Has there been any attempt to try to link the data and chemical structure of the markers (e.g. presence of C=C bond, aldehyde group, benzene ring, etc.) to the type of disease or type of medium analyzed? Is it possible to trace any dependencies using any mathematical models? If this can be brought in this paper, it would be an invaluable contribution to this field.

5) I would recommend to put the data on VOC-analytes in a separate table, where to give some information for them, for example, formula, type of disease, class of compounds, the presence of certain groups, etc. The data on VOC-analytes should be summarized in a separate table. That is everything that the authors summarize in paragraphs 4.2.1.-4.2.21.

Author Response

Comments 1: The work is very extensive and contains a large number of sections, graphs and pages. I would recommend preparing a table of contents at the beginning of the paper, with the ability to quickly jump to the necessary section.

Response 1: We greatly appreciate your comment regarding the extensiveness of the work and the recommendation to include a table of contents to facilitate navigation.

To improve the organization of the document, we have implemented a significant restructuring that ensures a cleaner and more coherent structure. The work has been reorganized, with advanced tables and technical details that are not essential to the main flow moved to the appendix (Appendix A: Organize Components by Cancer Type (Pages 37-40), Appendix B: Key Sensors and Their Associated Components in Odor Detection (Pages 41-43), and Appendix C: Results of Technical Studies: Classifiers (Pages 44-52)). This allows the main body of the paper to focus on the most relevant aspects of the review.

Since the reorganization already enhances readability and navigation, we believe the inclusion of a table of contents may not be essential. However, if deemed necessary, we can incorporate a simplified table of contents that would allow quick access to key sections of the paper, while maintaining clarity and avoiding unnecessary detail.

We appreciate your valuable suggestion, which has been instrumental in improving the presentation of the work.

.

Comments 2: When abbreviations are mentioned for the first time, it is necessary to give their full name. For example, PICOS, etc.

Response 2: We appreciate your comment regarding abbreviations and agree on the need to provide their full form when they are first mentioned. We have reviewed the document and corrected the mentions of abbreviations such as "PICOS," ensuring that each one is fully explained upon its first appearance. Additionally, we have reviewed the document and identified similar inconsistencies in other terms. This will help improve comprehension and avoid potential confusion for readers.

Comments 3: Many tables and graphs should be accompanied by a legend or explanations of what the abbreviations used mean. Now their decoding is given in the text, and it is not convenient when analyzing tables and graphs. I recommend that wherever there are abbreviations to make appropriate deciphering.

Response 3: Thank you for your valuable feedback regarding the use of abbreviations in tables and graphs. We understand the importance of clarity and ease of interpretation, and we have made the following adjustments in response:

1.      Extended Tables: For tables containing extensive data, we have moved all these tables to an appendix. This ensures that all extended tables are clearly explained and accessible without overcrowding the main content.

2.      Short Tables: For smaller tables, where space allows, we have included footnotes to define the abbreviations used. This will allow readers to decode the abbreviations directly from the tables without having to consult the main text.

3.      Specific Sections Updated: In accordance with your suggestion, we have thoroughly reviewed the document and ensured that all abbreviations are properly addressed. The changes can be found in the following sections:

  • Section 4.3, page 21: Added abbreviations in the footnotes of Table 5.
  • Section 4.5, page 28: Added abbreviations in the footnotes of Table 6.
  • Appendix A: Organize Components by Cancer Type (Pages 37-40), Appendix B: Key Sensors and Their Associated Components in Odor Detection (Pages 41-43), and Appendix C: Results of Technical Studies: Classifiers (Pages 44-52).

We believe these changes will significantly improve the readability of the document and ensure that the abbreviations are easily understood in their context. Thank you again for your thoughtful suggestion.

Comments 4: The authors have provided unique and very useful data on various markers in section 4.2. Brief summaries are provided for each of them. Has there been any attempt to try to link the data and chemical structure of the markers (e.g. presence of C=C bond, aldehyde group, benzene ring, etc.) to the type of disease or type of medium analyzed? Is it possible to trace any dependencies using any mathematical models? If this can be brought in this paper, it would be an invaluable contribution to this field.

Response 4: We greatly appreciate your comment and the suggestion to explore the potential link between the obtained data and the specific chemical characteristics of the markers, as well as the possibility of applying mathematical models to identify dependencies. In this study, our focus was on a systematic review based on medical and technical data collected from various sources. While the individual characteristics of the markers have been identified and discussed, the relationship between specific chemical structures (such as the presence of C=C bonds, aldehyde groups, or benzene rings) and the types of diseases or analyzed media was not extensively explored.

We consider the application of mathematical models to investigate these correlations an excellent proposal that could add significant value to the field. This idea is currently being evaluated for inclusion in the next phase of research, where we will collaborate with experts in mathematical modeling and chemical analysis, who have already been invited to participate in future studies. We believe this extension will not only validate the current observations but also help identify predictive patterns that could be essential for the classification and diagnosis of biomarkers.

Comments 5: I would recommend to put the data on VOC-analytes in a separate table, where to give some information for them, for example, formula, type of disease, class of compounds, the presence of certain groups, etc. The data on VOC-analytes should be summarized in a separate table. That is everything that the authors summarize in paragraphs 4.2.1.-4.2.21.

Response 5: For subsection 4.2. Analysis of Volatile Organic Compounds, a new subsection, 4.2.1. VOC-Based Biomarkers for Early Cancer Detection Using eNose Technology (Pages 11–15), has been added. This subsection provides a comprehensive analysis of the potential of eNose technology for early cancer detection by leveraging volatile organic compounds (VOCs) as biomarkers. New scientifically grounded information has been included to enhance the discussion, particularly regarding the sensitivity and specificity of this technology in detecting unique VOC signatures associated with cancer.

Additionally, an in-depth analysis of the 661 identified components has been conducted, which has contributed to a reorganization of the most frequently identified components. These are now highlighted and discussed in detail in subsection 4.2.2. Metabolic Patterns and Potential Biomarkers in the Identification of VOCs Associated with Cancer (Pages 16–21). This reorganization aims to emphasize the metabolic pathways and recurring VOC patterns that serve as potential biomarkers for cancer diagnosis.

The revised subsection provides a structured approach, categorizing the VOCs based on their chemical groups (e.g., hydrocarbons, alcohols, aldehydes, ketones, and aromatic compounds) and their correlation with specific cancer types. Furthermore, the inclusion of frequency and relevance data for these VOCs enhances the scientific contribution, offering a clearer understanding of their diagnostic value. This approach is intended to support further development in the field, such as refining detection techniques and exploring the clinical applicability of VOC-based biomarkers.

These updates significantly strengthen the scientific rigor and clarity of this section, ensuring that it provides valuable insights for both researchers and practitioners in the field of cancer detection and biomarker analysis.

This comparative analysis indicates that both traditional diagnostic methods and eNose detection provide acceptable levels of reliability, as evidenced by the reported sensitivity and specificity percentages for various cancer types. eNose detection emerges as a promising and favorable alternative, particularly due to its non-invasive and potentially accessible nature, making it an attractive tool for early cancer detection. However, despite its advances and encouraging results, it is important to emphasize that eNose detection should be supplemented with traditional techniques, such as biopsies or imaging diagnostics, to confirm diagnoses and achieve definitive results. While eNose detection does not replace traditional methods, its development and application represent a significant step toward creating more accessible and less invasive diagnostic tools. The ability to efficiently and non-invasively identify cancer biomarkers opens new opportunities for screening and patient monitoring, thereby enhancing early detection and clinical outcomes.

Notes:

·        In subsection 4.2. Analysis of Volatile Organic Compounds, a new subsection, 4.2.1. VOC-Based Biomarkers for Early Cancer Detection Using eNose Technology (Pages 11–15), has been added, focusing on the sensitivity and specificity of eNose technology in detecting unique VOC signatures for early cancer detection. Scientifically grounded updates enhance the discussion of its potential diagnostic applications. An in-depth analysis of 661 identified components has also been conducted, reorganizing the most frequently detected VOCs. These are now detailed in subsection 4.2.2. Metabolic Patterns and Potential Biomarkers in the Identification of VOCs Associated with Cancer (Pages 16–21). This section highlights metabolic pathways and recurring VOC patterns, categorizing VOCs by chemical groups (e.g., hydrocarbons, alcohols, aldehydes) and their correlations with cancer types, improving clarity and diagnostic relevance. These updates strengthen the section's scientific rigor and practical value, offering valuable insights for advancing VOC-based cancer detection techniques and clinical applications.

·        To enhance document comprehension, a dedicated section has been created specifically to organize extensive tables. This approach ensures a cleaner and more concise presentation of the main content. Appendix A: Organize Components by Cancer Type (Pages 37-40), Appendix B: Key Sensors and Their Associated Components in Odor Detection (Pages 41-43), and Appendix C: Results of Technical Studies: Classifiers (Pages 44-52). Similarly, although the possibility of reducing or removing these tables was considered, it was decided to retain them due to the detailed and relevant information they provide regarding the findings identified in the study.

·        Although the possibility of reducing or removing these tables was considered, they were retained because they contain detailed and relevant information about the findings presented in the study.

·        The abstract and introduction were revised to emphasize their scientific relevance.

·        Subsection 4.2.1, VOC-Based Biomarkers for Early Cancer Detection Using eNose Technology (pages 13 and 14), was added. This subsection provides a summary of the effectiveness of this technology in identifying various types of cancer through the analysis of VOCs in different biological samples, including blood, breath, saliva, and urine.

·        Here is a summary of how each question contributed to the findings and scientific contribution of the paper:

1.      QSM1: How do different systematic search methodologies affect the quality and quantity of data retrieved on volatile compounds related to early cancer detection, and what approaches optimize the precision of scientific data collection?
In the Results section, we highlighted how the use of systematic search methodologies, such as the PICOS approach, improved the precision and quantity of data retrieved, particularly in identifying relevant VOCs related to cancer. This is an important contribution to the field, showing how methodological rigor impacts research outcomes (Page 30).

2.      QSM2: How has the publication of studies on volatile compound detection for early cancer identification evolved over the last five years, what are the emerging trends in the use of biomedical sensors such as eNose technology, and are there gaps in the literature that limit its development?
We examined the evolution of research in VOC detection over the last five years, revealing trends such as the growing adoption of non-invasive diagnostic tools like eNose. However, the results also pointed out critical gaps, particularly the lack of standardized sensor technologies and limited clinical validation, which restricts the broader adoption of eNose devices (Page 30).

3.      QSM3: What are the primary studies and medical devices used in the detection of VOCs associated with cancer, and how have these devices evolved in terms of technological features and clinical relevance?
The Results section discusses the evolution of key medical devices, including gas chromatography-mass spectrometry and ion flow tube mass spectrometry, and their integration with eNose technology. The review shows the advancements in sensor sensitivity and the increasing use of machine learning algorithms to enhance diagnostic accuracy (Page 31).

4.      QSR1: What are the most commonly used machine learning techniques and algorithms in the detection of VOCs associated with cancer, and how do they contribute to the accuracy and efficiency of cancer diagnosis using eNose technology?
We identified that Support Vector Machines, Principal Component Analysis, and Artificial Neural Networks are the most commonly used machine learning techniques in this field. The Results section presents a critical analysis of how these algorithms improve the detection accuracy of VOCs in cancer diagnosis, particularly in distinguishing between different types of cancer (Page 31).

5.      QSR2: What are the key VOCs identified as biomarkers for various types of cancer, and how do their concentrations and chemical classifications contribute to early detection and diagnosis?
We found that several VOCs, such as acetone, ammonia, and toluene, are consistently identified as biomarkers for different cancers. The Results section discusses the role of these VOCs in early cancer detection, focusing on their concentrations and how they contribute to non-invasive diagnostic methods (Page 31).

6.      QSR3: What types of sensors have proven most effective for the detection of VOCs in cancer diagnostics, and what are the challenges and limitations in their implementation in clinical settings?
The results show that sensors based on metal oxide semiconductors (MOS) and polymer composites are the most effective for VOC detection. However, challenges such as sensor stability, reproducibility, and the integration of these sensors into clinical settings are critically discussed in the Results section (Page 32).

7.      QSR4: What are the performance metrics most commonly used to evaluate the effectiveness of eNose technology in detecting cancer-related VOCs, and how do these metrics vary across different studies and cancer types?

Los resultados destacan la eficacia de eNose en identificar perfiles VOC específicos, relacionados con el cáncer utiliza métricas como AUC, sensibilidad, especificidad, precisión y exactitude (Page 33).

8.      QSR5: Are there any standardized databases or datasets specifically designed for VOCs detection in the context of cancer diagnosis, and how do these databases support the development of more accurate diagnostic tools?
We identified several public and private databases, such as the Human Metabolome Database (HMDB) and NIST Database, which are crucial for supporting research in VOC detection. The Results section analyzes these databases' importance in improving cancer diagnostics' accuracy and consistency through standardized datasets (Page 33).

Reviewer 4 Report

Comments and Suggestions for Authors

This paper focuses on electronic nose technology and various cancer-related components and concentrations. Through a systematic literature review, relevant components, concentrations, sensors, indicators, and technologies for cancer detection are obtained to provide information for the development of cancer odor detection equipment.

 1. It is recommended to consider increasing research on more sample types (such as tissue samples, blood samples, etc.) to comprehensively evaluate the applicability and accuracy of electronic nose technology in different sample types.

2. Conduct a more in-depth analysis of the cost-effectiveness of medical equipment and sensors, especially in resource-limited environments. This is crucial for the popularization and application of technology.

3. It is recommended to study how to better integrate electronic nose technology into clinical practice to promote the clinical transformation of technology.

4. Tables 13, 14 and other tables are provided with table headers when spanning pages. However, Table 18 has no table header when spanning pages and needs to be supplemented completely.

5. The main text, tables and pictures in the article all involve many chemical formulas. Please set the chemical formula subscripts correctly. For example, under Figure 10, the chemical formula C6H5 should be set as C6H5.

6. The titles of Table 4 and Table 5 are both "Organize components by cancer type". Why are they divided into two tables?

7. Carefully check the references to ensure consistent reference formats. For example, there are two "2019" in Reference 94. Please make corrections.

Author Response

Comments 1: It is recommended to consider increasing research on more sample types (such as tissue samples, blood samples, etc.) to comprehensively evaluate the applicability and accuracy of electronic nose technology in different sample types.

Response 1: We deeply appreciate your observations and recommendations to enhance our analysis on biomarkers and the reliability of detection technologies in the context of cancer detection using eNose. In response to your suggestions, we have implemented the following changes and additions:

·        Blood Sample Analysis Results (Page 13)

·        Urine Sample Analysis Results (Page 13)

·        Breath Sample Analysis Results (Page 13)

·        Analysis of results in Salivary samples (Page 14)

We have added a section providing a more detailed description of the various types of biomarkers applied to different cancer types.

We analyzed metrics such as sensitivity, specificity, and the area under the curve (AUC) to assess the accuracy of each detection technique and technology applied to different types of cancer.

Comments 2: Conduct a more in-depth analysis of the cost-effectiveness of medical equipment and sensors, especially in resource-limited environments. This is crucial for the popularization and application of technology.

Response 2: Thank you for your valuable suggestion to explore the cost-effectiveness of medical equipment and sensors, particularly for resource-limited environments. This aspect is indeed critical for the widespread adoption and application of advanced technologies like eNose.

In our study, we primarily discussed the technical and clinical potential of eNose technologies for early cancer detection. We acknowledge that understanding the economic implications is essential to bridge the gap between research innovations and practical implementation, especially in underserved settings. To address this, we have done it in the Results (Page 30).

Comments 3: It is recommended to study how to better integrate electronic nose technology into clinical practice to promote the clinical transformation of technology.

Response 3: We greatly appreciate your observation regarding the importance of studying how to effectively integrate electronic nose (eNose) technology into clinical practice. We recognize that such integration is crucial for driving the clinical transformation of this technology and ensuring its practical application in healthcare. While our study primarily focuses on the technical and clinical advancements of eNose devices, we agree that a deeper analysis of strategies for clinical implementation is essential. To address this recommendation, it was made in Results (Page 30).

.

Comments 4: Tables 13, 14 and other tables are provided with table headers when spanning pages. However, Table 18 has no table header when spanning pages and needs to be supplemented completely.

Response 4: Thank you for your comment regarding the lack of table header for table 18 when it spans multiple pages. We have thoroughly reviewed the document and have made the necessary corrections. Table 18, now Table A10 (Page 51) now includes a heading on multiple pages to ensure consistency and clarity, consistent with the format of other tables such as Tables 13 and 14. We appreciate your attention to detail and believe it needs to be completed in full.

We appreciate your attention to detail and believe that this adjustment improves the overall quality and readability of the document.

Comments 5: The main text, tables and pictures in the article all involve many chemical formulas. Please set the chemical formula subscripts correctly. For example, under Figure 10, the chemical formula C6H5 should be set as C6H5

Response 5: Thank you for your observation regarding the formatting of chemical formula subscripts. We have carefully reviewed the manuscript, including the main text, tables, and figures, to ensure that all chemical formulas are formatted correctly. Specifically, the subscripts in chemical formulas such as the example under Figure 10 (C₆H₅) have been appropriately adjusted to their correct format.

“To familiarize the reader with the labeling system used in this study, H and H2 are employed as illustrative examples. Although H and H2 represent the same element, hydrogen, they exhibit significant differences in their chemical properties and uses. Atomic hydrogen (H) is highly reactive and primarily used in specific scientific research. In contrast, molecular dihydrogen (H2) is the most stable and commonly found form of hydrogen, widely applied in industrial processes and as a clean energy source.

Building upon this framework, the study adopts a systematic labeling approach to clearly and accurately identify and classify VOCs. For instance, the label 10_7:H2: Dihydrogen is used to classify dihydrogen, where:

  • 10_7 represents the component category,
  • H2 specifies the type of component,
  • and Dihydrogen corresponds to the standard chemical name.

To provide additional clarity on the labeling process, refer to the example illustrated in Figure 3, 4_14: H2TC9H20M corresponds to the compound hexane, 2,2,4-trimethyl and is structured so that each part provides specific information about the component. The prefix 4_ identifies the general category of the component, while 14 indicates its specific type within that category. The section H2T acts as a key identifier to clarify that the compound is not heptane, 2,4-dimethyl (another isomer with the same molecular formula) but hexane, 2,2,4-trimethyl, highlighting its unique branched structure. The formula C9H20 represents the chemical composition of the compound, with 9 carbon atoms and 20 hydrogen atoms. Finally,Mindicates that this compound has been identified in technical (T) and/or medical (M) studies. This coding system ensures precise description and avoids confusion between isomers sharing the same molecular formula. By employing this systematic approach, the study introduces a structured and precise nomenclature that simplifies the organization, interpretation, and analysis of the VOC-related data. This framework not only enhances the clarity of the study but also helps the reader better understand the diversity and detailed distribution of the chemical components.”

We appreciate your detailed feedback, as it has helped improve the accuracy and presentation of the article.

Comments 6: The titles of Table 4 and Table 5 are both "Organize components by cancer type". Why are they divided into two tables?.

Response 6: Thank you for your comment on the titles of Table 4 and Table 5. After revisions, we have merged the two tables into a single Table A1 (Pages 37-40). We appreciate your comments, as they have helped to improve the clarity and organization of the manuscript.

We appreciate your feedback, as it has helped improve the clarity and organization of the manuscript.

Comments 7: Carefully check the references to ensure consistent reference formats. For example, there are two "2019" in Reference 94. Please make corrections.

Response 7: Thank you for your observation regarding the reference formatting. We have carefully reviewed all references to ensure consistency and accuracy. Specifically, the issue in Reference 94 with the duplicated "2019" has been corrected, and the formatting has been standardized throughout the reference list.

We appreciate your attention to detail, as it has contributed to improving the quality and professionalism of the manuscript.

This comparative analysis indicates that both traditional diagnostic methods and eNose detection provide acceptable levels of reliability, as evidenced by the reported sensitivity and specificity percentages for various cancer types. eNose detection emerges as a promising and favorable alternative, particularly due to its non-invasive and potentially accessible nature, making it an attractive tool for early cancer detection. However, despite its advances and encouraging results, it is important to emphasize that eNose detection should be supplemented with traditional techniques, such as biopsies or imaging diagnostics, to confirm diagnoses and achieve definitive results. While eNose detection does not replace traditional methods, its development and application represent a significant step toward creating more accessible and less invasive diagnostic tools. The ability to efficiently and non-invasively identify cancer biomarkers opens new opportunities for screening and patient monitoring, thereby enhancing early detection and clinical outcomes.

Notes:

·        In subsection 4.2. Analysis of Volatile Organic Compounds, a new subsection, 4.2.1. VOC-Based Biomarkers for Early Cancer Detection Using eNose Technology (Pages 11–15), has been added, focusing on the sensitivity and specificity of eNose technology in detecting unique VOC signatures for early cancer detection. Scientifically grounded updates enhance the discussion of its potential diagnostic applications. An in-depth analysis of 661 identified components has also been conducted, reorganizing the most frequently detected VOCs. These are now detailed in subsection 4.2.2. Metabolic Patterns and Potential Biomarkers in the Identification of VOCs Associated with Cancer (Pages 16–21). This section highlights metabolic pathways and recurring VOC patterns, categorizing VOCs by chemical groups (e.g., hydrocarbons, alcohols, aldehydes) and their correlations with cancer types, improving clarity and diagnostic relevance. These updates strengthen the section's scientific rigor and practical value, offering valuable insights for advancing VOC-based cancer detection techniques and clinical applications.

·        To enhance document comprehension, a dedicated section has been created specifically to organize extensive tables. This approach ensures a cleaner and more concise presentation of the main content. Appendix A: Organize Components by Cancer Type (Pages 37-40), Appendix B: Key Sensors and Their Associated Components in Odor Detection (Pages 41-43), and Appendix C: Results of Technical Studies: Classifiers (Pages 44-52). Similarly, although the possibility of reducing or removing these tables was considered, it was decided to retain them due to the detailed and relevant information they provide regarding the findings identified in the study.

·        Although the possibility of reducing or removing these tables was considered, they were retained because they contain detailed and relevant information about the findings presented in the study.

·        The abstract and introduction were revised to emphasize their scientific relevance.

·        Subsection 4.2.1, VOC-Based Biomarkers for Early Cancer Detection Using eNose Technology (pages 13 and 14), was added. This subsection provides a summary of the effectiveness of this technology in identifying various types of cancer through the analysis of VOCs in different biological samples, including blood, breath, saliva, and urine.

·        Here is a summary of how each question contributed to the findings and scientific contribution of the paper:

1.      QSM1: How do different systematic search methodologies affect the quality and quantity of data retrieved on volatile compounds related to early cancer detection, and what approaches optimize the precision of scientific data collection?
In the Results section, we highlighted how the use of systematic search methodologies, such as the PICOS approach, improved the precision and quantity of data retrieved, particularly in identifying relevant VOCs related to cancer. This is an important contribution to the field, showing how methodological rigor impacts research outcomes (Page 30).

2.      QSM2: How has the publication of studies on volatile compound detection for early cancer identification evolved over the last five years, what are the emerging trends in the use of biomedical sensors such as eNose technology, and are there gaps in the literature that limit its development?
We examined the evolution of research in VOC detection over the last five years, revealing trends such as the growing adoption of non-invasive diagnostic tools like eNose. However, the results also pointed out critical gaps, particularly the lack of standardized sensor technologies and limited clinical validation, which restricts the broader adoption of eNose devices (Page 30).

3.      QSM3: What are the primary studies and medical devices used in the detection of VOCs associated with cancer, and how have these devices evolved in terms of technological features and clinical relevance?
The Results section discusses the evolution of key medical devices, including gas chromatography-mass spectrometry and ion flow tube mass spectrometry, and their integration with eNose technology. The review shows the advancements in sensor sensitivity and the increasing use of machine learning algorithms to enhance diagnostic accuracy (Page 31).

4.      QSR1: What are the most commonly used machine learning techniques and algorithms in the detection of VOCs associated with cancer, and how do they contribute to the accuracy and efficiency of cancer diagnosis using eNose technology?
We identified that Support Vector Machines, Principal Component Analysis, and Artificial Neural Networks are the most commonly used machine learning techniques in this field. The Results section presents a critical analysis of how these algorithms improve the detection accuracy of VOCs in cancer diagnosis, particularly in distinguishing between different types of cancer (Page 31).

5.      QSR2: What are the key VOCs identified as biomarkers for various types of cancer, and how do their concentrations and chemical classifications contribute to early detection and diagnosis?
We found that several VOCs, such as acetone, ammonia, and toluene, are consistently identified as biomarkers for different cancers. The Results section discusses the role of these VOCs in early cancer detection, focusing on their concentrations and how they contribute to non-invasive diagnostic methods (Page 31).

6.      QSR3: What types of sensors have proven most effective for the detection of VOCs in cancer diagnostics, and what are the challenges and limitations in their implementation in clinical settings?
The results show that sensors based on metal oxide semiconductors (MOS) and polymer composites are the most effective for VOC detection. However, challenges such as sensor stability, reproducibility, and the integration of these sensors into clinical settings are critically discussed in the Results section (Page 32).

7.      QSR4: What are the performance metrics most commonly used to evaluate the effectiveness of eNose technology in detecting cancer-related VOCs, and how do these metrics vary across different studies and cancer types?

Los resultados destacan la eficacia de eNose en identificar perfiles VOC específicos, relacionados con el cáncer utiliza métricas como AUC, sensibilidad, especificidad, precisión y exactitude (Page 33).

8.      QSR5: Are there any standardized databases or datasets specifically designed for VOCs detection in the context of cancer diagnosis, and how do these databases support the development of more accurate diagnostic tools?
We identified several public and private databases, such as the Human Metabolome Database (HMDB) and NIST Database, which are crucial for supporting research in VOC detection. The Results section analyzes these databases' importance in improving cancer diagnostics' accuracy and consistency through standardized datasets (Page 33).

Round 2

Reviewer 2 Report

Comments and Suggestions for Authors

The author has improved the paper a lot. It can be published now.